# Efficient inference and identifiability analysis for differential equation models with random parameters

**Alexander P. Browning**[1,2,3], **Christopher Drovandi**[1,2], **Ian W. Turner**[1], **Adrianne L. Jenner**[1,2], **Matthew J. Simpson**[1,2]*

**1** School of Mathematical Sciences, Queensland University of Technology, Brisbane, Australia, **2** QUT Centre for Data Science, Queensland University of Technology, Brisbane, Australia, **3** Mathematical Institute, University of Oxford, Oxford, United Kingdom

* matthew.simpson@qut.edu.au

**Data Availability Statement:** All relevant data are within the manuscript and its Supporting information files.

## Abstract

Heterogeneity is a dominant factor in the behaviour of many biological processes. Despite this, it is common for mathematical and statistical analyses to ignore biological heterogeneity as a source of variability in experimental data. Therefore, methods for exploring the identifiability of models that explicitly incorporate heterogeneity through variability in model parameters are relatively underdeveloped. We develop a new likelihood-based framework, based on moment matching, for inference and identifiability analysis of differential equation models that capture biological heterogeneity through parameters that vary according to probability distributions. As our novel method is based on an approximate likelihood function, it is highly flexible; we demonstrate identifiability analysis using both a frequentist approach based on profile likelihood, and a Bayesian approach based on Markov-chain Monte Carlo. Through three case studies, we demonstrate our method by providing a didactic guide to inference and identifiability analysis of hyperparameters that relate to the statistical moments of model parameters from independent observed data. Our approach has a computational cost comparable to analysis of models that neglect heterogeneity, a significant improvement over many existing alternatives. We demonstrate how analysis of random parameter models can aid better understanding of the sources of heterogeneity from biological data.

## Author summary

Heterogeneity is a dominant factor in the behaviour of many biological and biophysical processes, and is often a primary source of the variability evident in experimental data. Despite this, it is relatively rare for mathematical models of biological systems to incorporate variability in model parameters as a source of noise. Therefore, methods for analysing whether model parameters and sources of variability are identifiable from commonly reported experimental data are relatively underdeveloped. As we demonstrate, such identifiability analysis is vital for model selection, experimental design, and gaining biological

**Funding:** This work is supported by the Australian Research Council (ARC) https://www.arc.gov.au/ through a Discovery Grant to MJS (Grant number DP200100177) and a Future Fellowship to CD (Grant number FT210100260). The funders had no role in study design, data collection and analysis, decision to publish, or preparation of the manuscript.

**Competing interests:** The authors have declared that no competing interests exist.

insights. In this work, we develop a fast, approximate framework for model calibration and identifiability analysis of mathematical models that incorporate biological heterogeneity through random parameters. Our method is highly flexible, and can be employed in both frequentist and Bayesian inference paradigms. Compared to alternative approaches, our approach is computationally efficient, with a computational cost comparable to analysis of standard models that neglect parameter variability.

This is a *PLOS Computational Biology* Methods paper.

## Introduction

Heterogeneity is understood as a dominant factor in the behaviour of many biological and biophysical systems [1–3]. Mathematical analysis of these systems is often constrained to parameter-fitting of differential equation based models. In many cases, heterogeneity is neglected, with variability in the data assumed to be noise and incorporated through independent, probabilistic observation processes [4–8].

Mathematical models have long been an essential tool for understanding the behaviour of systems from quantitative and experimental data. Parameter estimation allows practitioners to quantify observed behaviour in terms of parameters that carry physical interpretations. Establishing whether model parameters can be *identified* from the quantity and quality of experimental data available is critical for tailoring model and data complexity to the scientific questions of interest [7, 9–12]. Furthermore, predictions from non-identifiable models can be unreliable [9]. Such identifiability analysis is well established for ordinary differential equation models [9, 10, 13], stochastic models [12, 14–16], and, recently, partial differential equation models [8, 17]. There is, however, comparatively little guidance for identifiability analysis for models that explicitly incorporate heterogeneity in model parameters, limiting the ability of practitioners to identify and predict sources of biological variability.

In biological systems, heterogeneity might arise due to inter-experiment variability, gene expression [18], or patient-to-patient variability [19]. Even from tightly controlled experiments is data variability evident, potentially due to differences in cell behaviour between experiments. We demonstrate this in Fig 1A by showing results of an *in vitro* multicellular tumour spheroid experiment, using melanoma tumour spheroids generated from a single cell line and imaged using microscopy after seven days of growth [20]. Despite similarities in both size and morphology, spheroids are not identical. In Fig 1B, we summarise the experimental images with the most obvious measurement corresponding to the radius of a circle with the same cross-sectional area, and repeat this for ten spheroids collected from eight observation days (yielding 80 independent measurements). We also show predictions from a calibrated logistic model [21, 22], with a prediction interval capturing variability in the data through additive normal noise, a common assumption [9]. Key questions posed by the data in Fig 1 might relate to whether variability observed in the data is due to heterogeneity in the initial condition (spheroids are seeded with *approximately* 5000 cells), due to heterogeneity in the dynamic behaviour (differences in experimental conditions such as the concentration of nutrient might yield differences in the growth rate between spheroids), or due to extrinsic factors, including measurement noise.

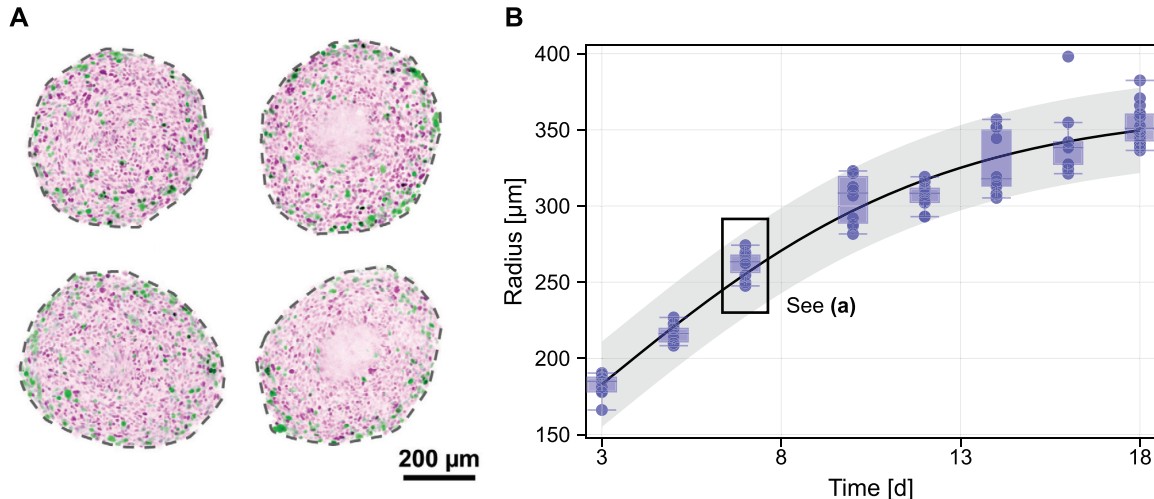

**Fig 1. Heterogeneity in experimental measurements of spheroid growth.** (A) Microscopy images of tumour spheroids grown from melanoma cell line WM983b from which a subset of measurements in (B) were taken. Spheroids were grown from 5000 cells, harvested, fixed, and imaged after nine days. Cells are transduced with fluorescent cell cycle indicators: colouring indicates cells in gap 1 (purple) and cells in gap 2 (green). In all but one spheroid, a lack of definition in the spheroid centre indicates the presence of a necrotic core. (B) Images are summarised by the radius of a circle with equivalent cross-sectional area as microscopy images. Shown are experimental measurements (blue discs and box plots) and a calibrated logistic model with prediction interval based on additive normal noise. Data from [20].

Differential equation models are widely used throughout biology, and have the potential to capture heterogeneity by relaxing the requirement that model parameters be fixed between measurements [23]. In the mathematical literature, such models are termed heterogeneous ordinary differential equations (ODEs) [24], random ODEs [25], or populations of models [26]. In the statistical literature, ODE-constrained hierarchical models, random-effects models, or non-linear mixed-effects models describe heterogeneity through a parameter hierarchy where model parameters have specified distributions that are themselves described by hyper-parameters [23, 27–30]. The literature is further split by a distinction between inference proce-dures that assume distributional forms [23, 28–33] or those that do not [24, 34], when inferring the underlying parameter distributions from quantitative observations. For the non-linear models common to biology, inference for random parameter variants is often computa-tionally costly—with cost that can increase significantly with the data sample size—and requires non-trivial selection of tuning parameters. Furthermore, there is very little guidance in the literature for assessing the identifiability, and hence ability of practitioners to determine the source of variability or even the benefit of considering parameter variability, using these classes of random parameter models.

Motivated by these observations, we develop a novel, approximate, and computationally efficient likelihood-based approach to inference and identifiability analysis of differential equation based models with random parameters based on an approximate moment-matched solution to the random parameter model [35]. To do this, we express the solution to the model about the parameter mean using a second order Taylor series expansion, which can be manip-ulated to obtain approximate expressions for the first three statistical moments of the model output distributions in terms of the statistical moments of the input (i.e., parameter) distribu-tions. Our method may therefore be classified as a moment-matching method similar to meth-ods routinely employed for stochastic fixed-parameters models such as the linear-noise approximation [36–39]. We highlight that, although we assume a distributional form for the

input parameter distributions, it is the statistical moments of the parameters that are inferred. Given the challenges of formulating high-dimensional distributions with possibly highly non-linear dependence structures, we focus our approach to low-dimensional data; for example, univariate or bivariate measurements collected independently at observation times, common in biology due to the challenges in collecting data and where samples are often destroyed for data to be collected [40–42]. While not our focus, our method is applicable to data of any dimension (including, for example, time-series data) provided that dependent measurements are approximately multivariate normally distributed, or can be transformed to meet this requirement. The restriction is less strict for univariate and bivariate data, where we are also able to capture the skewness in the data. By leveraging techniques such as automatic differentiation to construct the Taylor series expansion, our approach provides a deterministic approximation to the data likelihood with comparable computational cost to the corresponding fixed-parameter model.

Our approach differs from many as we construct a surrogate likelihood directly from the approximate distributional solution to the random parameter model, alleviating the need to either infer individual-level parameters or marginalise the posterior in non-linear mixed-effects modelling [30] and Bayesian hierarchical modelling [29], respectively. The availability of a surrogate likelihood allows us to perform inference and identifiability analysis of random parameter models using the standard suite of tools, including profile likelihood [9], Fisher information [43], and Markov-chain Monte-Carlo [7]. Aside from assessing the identifiability of hyperparameters—parameters that relate to the distributional form of the random parameters in the dynamical model—we demonstrate our method by answering a number of questions specific to identifiability analysis of random parameter models. Namely, whether variation in model parameters can be distinguished from measurement noise; whether identifiability of a random-parameter model differs from that of a fixed-parameter model; and, finally, how the order of the moment-matching approximation affects parameter identifiability. To aid in the uptake of both random parameter models and their application to better interpret the variability ubiquitous to biological data, we provide a Julia module implementing our approach on Github.

## Methods

We consider ODE state-space models of the form

$$\frac{\mathrm{d}\boldsymbol{x}(t, \boldsymbol{\theta})}{\mathrm{d}t} = \boldsymbol{g}(t, \boldsymbol{x}(t, \boldsymbol{\theta}), \boldsymbol{\theta}), \qquad \boldsymbol{x}(0, \boldsymbol{\theta}) = \boldsymbol{x}_0(\boldsymbol{\theta}), \tag{1}$$

subject to the observation process

$$\boldsymbol{y}(t, \boldsymbol{\theta}) = \boldsymbol{h}(\boldsymbol{x}(t, \boldsymbol{\theta}), \boldsymbol{\theta}), \tag{2}$$

at times $t \in \{t_1, t_2, \ldots, t_n\} = \mathcal{T}$. Here, $\boldsymbol{x}(t, \boldsymbol{\theta}) : \mathbb{R} \times \mathbb{R}^d \to \mathbb{R}^p$ is the state, $\boldsymbol{g}(t, \boldsymbol{x}, \boldsymbol{\theta}) : \mathbb{R} \times \mathbb{R}^p \times \mathbb{R}^d \to \mathbb{R}^p$ is the time-derivative of the state and $\boldsymbol{\theta} \in \mathbb{R}^d$ is a vector of parameters, traditionally assumed to be fixed between measurements [5, 21]. The observation process, $\boldsymbol{y}(t, \boldsymbol{\theta}) : \mathbb{R} \times \mathbb{R}^d \to \mathbb{R}^q$, represents potentially incomplete observations of the underlying state, characterised by $\boldsymbol{h}(\boldsymbol{x}, \boldsymbol{\theta}) : \mathbb{R}^p \times \mathbb{R}^d \to \mathbb{R}^q$.

The focus of this work is random parameter dynamical models, that is, dynamical models where model parameters vary between observations such that $\boldsymbol{\theta}$ is a random variable. In distinction to other classes of random or stochastic differential equations, we emphasise that $\boldsymbol{\theta}$ does not depend on $t$. In this formulation, it is possible to incorporate a probabilistic observation process directly into Eq (2) (for example, normally distributed measurement error)

through a sequence of random parameters $\varepsilon_i$ contained in $\boldsymbol{\theta}$ associated with each observation time $t_i$. For instance, to model additive normal noise, we would set the $i$th component of the observation process to $h_i(\boldsymbol{x}(t_i, \boldsymbol{\theta}), \boldsymbol{\theta}) = \bar{h}_i(\boldsymbol{x}(t_i, \boldsymbol{\theta}), \boldsymbol{\theta}) + \varepsilon_i$ where $\varepsilon_i \sim \mathcal{N}(0, \sigma^2)$ captures noise and $\bar{h}(\cdot)$ represents a noiseless observation from the model. We demonstrate both additive and multiplicative normal noise in this work, and highlight the flexibility gained by incorporating measurement error directly into the observation process through an additional random parameter.

Therefore, the model can be considered as a transformation of the random variable $\boldsymbol{\theta}$ to randomly distributed observations, $\boldsymbol{y}$. We denote a vector of dependent measurements

$$\boldsymbol{f}(\boldsymbol{\theta}) = [f_1(\boldsymbol{\theta}), \dots, f_n(\boldsymbol{\theta})]^\mathsf{T}. \tag{3}$$

For time-series data, $\boldsymbol{f}$ may represent dependent observations taken from an entire dependent trace; for example, in the case of univariate observations at times $t_1, \dots, t_n$, we have $\boldsymbol{y}(t, \boldsymbol{\theta}) = y(t, \boldsymbol{\theta})$ and $\boldsymbol{f}(\boldsymbol{\theta}) = [y(t_1, \boldsymbol{\theta}), \dots, y(t_n, \boldsymbol{\theta})]^\mathsf{T}$. For multivariate observations, we concatenate these observations such that

$$\boldsymbol{f}(\boldsymbol{\theta}) = [\boldsymbol{y}^{(1)}(t_1, \boldsymbol{\theta}), \dots, \boldsymbol{y}^{(1)}(t_n, \boldsymbol{\theta}), \dots, \boldsymbol{y}^{(m)}(t_1, \boldsymbol{\theta}), \dots, \boldsymbol{y}^{(m)}(t_n, \boldsymbol{\theta})]^\mathsf{T}, \tag{4}$$

where $\boldsymbol{y}^{(k)}(t_i, \boldsymbol{\theta})$ denotes a measurement from the $k$th component of $\boldsymbol{y}(t_i, \boldsymbol{\theta})$. For a tumour spheroid experiment, this might represent time-series radius measurements (for univariate observations) or radius and inner structure measurements (for multivariate observations) from a single spheroid throughout the course of the experiment. Alternatively, for so-called *snapshot data*, where observations are taken at each observation time independently, we consider a series of transformations that can be handled independently, $\boldsymbol{f}^{(1)}(\boldsymbol{\theta}), \boldsymbol{f}^{(2)}(\boldsymbol{\theta}), \dots$ where $\boldsymbol{f}^{(i)}(\boldsymbol{\theta}) = \boldsymbol{y}(t_i; \boldsymbol{\theta})$. For a tumour spheroid experiment, $\boldsymbol{f}^{(i)}(\boldsymbol{\theta})$ might represent a radius measurement collected by terminating a single tumour spheroid experiment at time $t_i$. The key difference between time-series and snapshot data is that in the case of the former, data from all time points are considered a single, dependent, multivariate measurement for which the covariance structure must be considered; whereas for the latter, data from each time point can be considered entirely independently, significantly reducing the dimensionality of the problem.

## Approximate solution of the random parameter model

Only in very limited cases (specifically, where the inverse $\boldsymbol{f}^{-1}$ is tractable) can the density of $\boldsymbol{f}(\boldsymbol{\theta})$ be calculated directly. Indeed, of primary interest in our work is the case where independent observations at a series of observation times are collected where it is highly likely that there are fewer observations than random model parameters, so it is likely that $\boldsymbol{f}^{-1}$ will not exist. Therefore, we build an approximate surrogate likelihood based on a Taylor series approximation to the moments of $\boldsymbol{f}(\boldsymbol{\theta})$ given the moments of $\boldsymbol{\theta}$ under the assumption that $\boldsymbol{f}$ is sufficiently smooth.

First, consider a univariate transformation, $f(\theta) : \mathbb{R} \to \mathbb{R}$ of the random variable $\theta \in \mathbb{R}$. To formulate expressions describing the moments of $f(\theta)$ (and hence, an approximate expression for the density of $f(\theta)$), consider the Taylor expansion of $f(\theta)$ about $\theta = \hat{\theta}$,

$$f(\theta) = f(\hat{\theta}) + \frac{\mathrm{d}f(\hat{\theta})}{\mathrm{d}\theta}(\theta - \hat{\theta}) + \frac{1}{2}\frac{\mathrm{d}^2 f(\hat{\theta})}{\mathrm{d}\theta^2}(\theta - \hat{\theta})^2 + \cdots. \tag{5}$$

If we choose $\hat{\theta} = \mathbb{E}(\theta)$, then the expectation of Eq (5) yields an equation for $\mathbb{E}(f(\theta))$ to expressions relating to $\mathbb{E}(\theta - \hat{\theta}) = 0$ and $\mathbb{E}((\theta - \hat{\theta})^2) = \mathbb{V}(\theta)$ (the variance of $\theta$). Similarly, each

side of Eq (5) can be squared or raised to higher powers to obtain expressions relating to the variance and skewness of $f(\theta)$.

Eq (5) readily extends to transformations of multivariate random variables. For instance, consider now that $\boldsymbol{\theta} = [\theta_1, \theta_2, \ldots, \theta_d]^\top \in \mathbb{R}^d$ and that $\boldsymbol{f}(\boldsymbol{\theta}) = [f_1(\boldsymbol{\theta}), \ldots, f_n(\boldsymbol{\theta})]^\top$. An expression for the $i$th component, $f_i(\boldsymbol{\theta})$, can be expressed in the following form using a Taylor expansion around $\boldsymbol{\theta} = \hat{\boldsymbol{\theta}}$,

$$f_i(\boldsymbol{\theta}) = f_i(\hat{\boldsymbol{\theta}}) + \sum_{a=1}^{d} \frac{\partial f_i(\hat{\boldsymbol{\theta}})}{\partial \theta_a} (\theta_a - \hat{\theta}_a) + \frac{1}{2} \sum_{a=1}^{d} \sum_{b=1}^{d} \frac{\partial^2 f_i(\hat{\boldsymbol{\theta}})}{\partial \theta_a \partial \theta_b} (\theta_a - \hat{\theta}_a)(\theta_b - \hat{\theta}_b) + \cdots. \quad (6)$$

While expectations of Eq (6) can still be taken, it is now more difficult to relate terms to the central moments of $\boldsymbol{\theta}$, particularly when Eq (6) is raised to higher powers. However, this task becomes clearer when Eq (6) is expressed in matrix or tensor notation: the terms relating to the second derivatives in Eq (6), for example, are related to the Frobenius inner product (i.e., sum of the component-wise product of two matrices or tensors) of the Hessian matrix and a matrix that becomes the covariance matrix when expectations are taken. This notation yields

$$f_i(\boldsymbol{\theta}) = f_i(\hat{\boldsymbol{\theta}}) + \nabla f_i(\hat{\boldsymbol{\theta}}) \cdot (\boldsymbol{\theta} - \hat{\boldsymbol{\theta}}) + \frac{1}{2} H f_i(\hat{\boldsymbol{\theta}}) \circ M_2(\boldsymbol{\theta} - \hat{\boldsymbol{\theta}}) + \cdots. \quad (7)$$

Here, $\circ$ denotes the Frobenius inner product, $Hf_i(\boldsymbol{\theta})$ is the Hessian matrix of $f_i(\boldsymbol{\theta})$, i.e., a matrix with elements

$$[Hf_i(\hat{\boldsymbol{\theta}})]_{ab} = \frac{\partial^2 f_i(\hat{\boldsymbol{\theta}})}{\partial \theta_a \partial \theta_b}, \quad (8)$$

and $M_2$ is an operator that returns the matrix formed by taking an outer product of a vector with itself, with elements given by

$$[M_2(\boldsymbol{\theta} - \hat{\boldsymbol{\theta}})]_{ab} = (\theta_a - \hat{\theta}_a)(\theta_b - \hat{\theta}_b). \quad (9)$$

Similarly, higher order operators are defined by $M_3$, which returns a three-dimensional tensor, and $M_4$ which returns a four-dimensional tensor. We form $M_2$ using a generalisation of the Kronecker product, $M_2(\boldsymbol{\theta} - \hat{\boldsymbol{\theta}}) = (\boldsymbol{\theta} - \hat{\boldsymbol{\theta}}) \otimes (\boldsymbol{\theta} - \hat{\boldsymbol{\theta}})$ and $M_k(\boldsymbol{\theta} - \hat{\boldsymbol{\theta}}) = (\boldsymbol{\theta} - \hat{\boldsymbol{\theta}}) \otimes M_{k-1}(\boldsymbol{\theta} - \hat{\boldsymbol{\theta}})$, where $\otimes$ is defined such that, for two vectors $\mathbf{a}$ and $\mathbf{b}$,

$$\mathbf{a} \otimes \mathbf{b} = \begin{bmatrix} a_1 \mathbf{b}^\top \\ a_2 \mathbf{b}^\top \\ \vdots \\ a_n \mathbf{b}^\top \end{bmatrix}. \quad (10)$$

The operation is similarly defined for arguments in higher-dimensions, returning a tensor of dimensionality equal to the sum of the dimensions of both arguments. For brevity, we define a Kronecker power operator such that $\mathbf{a}^{\otimes n}$ refers to the operation performed on $\mathbf{a}$ by itself $n$ times.

Defining $\langle \cdot \rangle = \mathbb{E}(\cdot)$ and noting that, in our notation, $\mathbb{V}(\boldsymbol{\theta}) = \langle M_2(\boldsymbol{\theta} - \hat{\boldsymbol{\theta}}) \rangle$ (and similar for higher order moments relating to the skewness tensor, $\mathbb{S}$ and kurtosis tensor, $\mathbb{K}$), we can show

that

$$\langle f_i(\boldsymbol{\theta}) \rangle \approx f_i(\hat{\boldsymbol{\theta}}) + \mathbb{V}(\boldsymbol{\theta}) \circ \frac{1}{2} H f_i(\hat{\boldsymbol{\theta}}), \tag{11}$$

$$\begin{aligned}
\langle f_i^2(\boldsymbol{\theta}) \rangle &\approx f_i^2(\hat{\boldsymbol{\theta}}) + \mathbb{V}(\boldsymbol{\theta}) \circ \left( \nabla f_i(\hat{\boldsymbol{\theta}})^{\otimes 2} + f_i(\hat{\boldsymbol{\theta}}) H f_i(\hat{\boldsymbol{\theta}}) \right) \\
&\quad + \mathbb{S}(\boldsymbol{\theta}) \circ (H f_i(\hat{\boldsymbol{\theta}}) \otimes \nabla f_i(\hat{\boldsymbol{\theta}})) \\
&\quad + \mathbb{K}(\boldsymbol{\theta}) \circ \frac{1}{4} H f_i(\hat{\boldsymbol{\theta}})^{\otimes 2},
\end{aligned} \tag{12}$$

$$\begin{aligned}
\langle f_i^3(\boldsymbol{\theta}) \rangle &\approx f_i^3(\hat{\boldsymbol{\theta}}) + \mathbb{V}(\boldsymbol{\theta}) \circ \frac{3}{2} f_i(\hat{\boldsymbol{\theta}}) \left( 2 \nabla f_i(\hat{\boldsymbol{\theta}})^{\otimes 2} + f_i(\hat{\boldsymbol{\theta}}) H f_i(\hat{\boldsymbol{\theta}}) \right) \\
&\quad + \mathbb{S}(\boldsymbol{\theta}) \circ \left( \nabla f_i(\hat{\boldsymbol{\theta}})^{\otimes 3} + 3 f_i(\hat{\boldsymbol{\theta}}) H f_i(\hat{\boldsymbol{\theta}}) \otimes \nabla f_i(\hat{\boldsymbol{\theta}}) \right) \\
&\quad + \mathbb{K}(\boldsymbol{\theta}) \circ 3 \left( \frac{1}{4} f_i(\hat{\boldsymbol{\theta}}) H f_i(\hat{\boldsymbol{\theta}})^{\otimes 2} + \frac{1}{2} \nabla f_i(\hat{\boldsymbol{\theta}})^{\otimes 2} \otimes H f_i(\hat{\boldsymbol{\theta}}) \right).
\end{aligned} \tag{13}$$

and

$$\begin{aligned}
\langle f_i(\boldsymbol{\theta}) f_j(\boldsymbol{\theta}) \rangle &\approx f_i(\hat{\boldsymbol{\theta}}) f_j(\hat{\boldsymbol{\theta}}) \\
&\quad + \mathbb{V}(\boldsymbol{\theta}) \circ \frac{1}{2} \left( f_i(\hat{\boldsymbol{\theta}}) H f_j(\hat{\boldsymbol{\theta}}) + f_j(\hat{\boldsymbol{\theta}}) H f_i(\hat{\boldsymbol{\theta}}) + 2 \nabla f_i(\hat{\boldsymbol{\theta}}) \otimes f_j(\hat{\boldsymbol{\theta}}) \right) \\
&\quad + \mathbb{S}(\boldsymbol{\theta}) \circ \left( \nabla f_i(\hat{\boldsymbol{\theta}}) \otimes H f_j(\hat{\boldsymbol{\theta}}) + \nabla f_j(\hat{\boldsymbol{\theta}}) \otimes H f_i(\hat{\boldsymbol{\theta}}) \right) \\
&\quad + \mathbb{K}(\boldsymbol{\theta}) \circ \frac{1}{4} H f_i(\hat{\boldsymbol{\theta}}) \otimes H f_j(\hat{\boldsymbol{\theta}}).
\end{aligned} \tag{14}$$

Note that we have applied the closure $\langle M_k(\boldsymbol{\theta}) \rangle = \mathbf{0}$ for $k \geq 5$. Formal derivations of Eq (7) and Eqs (11) to (14) are provided as supporting material (S1 File).

Eqs (11) to (14) provide approximate expressions for the mean vector with entries $\boldsymbol{\mu}_i = \langle f_i(\boldsymbol{\theta}) \rangle$, covariance matrix with entries $\Sigma_{ij} = \langle f_i(\boldsymbol{\theta}) f_j(\boldsymbol{\theta}) \rangle - \langle f_i(\boldsymbol{\theta}) \rangle \langle f_j(\boldsymbol{\theta}) \rangle$ and univariate skewnesses vector with entries $\boldsymbol{\omega}_i = \langle (f_i(\boldsymbol{\theta}) - \boldsymbol{\mu}_i)^3 \rangle / \Sigma_{ij}^{3/2}$, of $\boldsymbol{f}(\boldsymbol{\theta})$. From this, we construct an approximate density function for $\boldsymbol{f}(\boldsymbol{\theta})$ using two approaches: one based on a multivariate normal distribution that matches the first two moments, and another based on a gamma distribution that matches the first three.

The *normal* or *two moment* approach approximates

$$\boldsymbol{f}(\boldsymbol{\theta}) \sim \text{MvNormal}(\boldsymbol{\mu}, \Sigma). \tag{15}$$

The primary advantage of this approach is that we can form approximations without regard to the dimensionality of $\boldsymbol{f}(\boldsymbol{\theta})$. Furthermore, it is overwhelmingly the case in the mathematical biology literature, for instance, that normality is assumed when calibrating dynamical models to experimental data [5, 8].

The *gamma* or *three moment* approach approximates marginal distributions with a shifted gamma distribution parameterised in terms of its mean, variance and skewness [32],

$$f_i(\boldsymbol{\theta}) \sim \text{ShiftedGamma}(\boldsymbol{\mu}_i, \Sigma_{ii}, \boldsymbol{\omega}_i). \tag{16}$$

This approach is advantageous as it recaptures the normal approach in the limit $\boldsymbol{\omega}_i \to 0$, but allows more flexibility in terms of the shape of the distribution. The primary difficulty of the gamma approach is to construct an approximation to multivariate $\boldsymbol{f}(\boldsymbol{\theta})$. We do this in the case

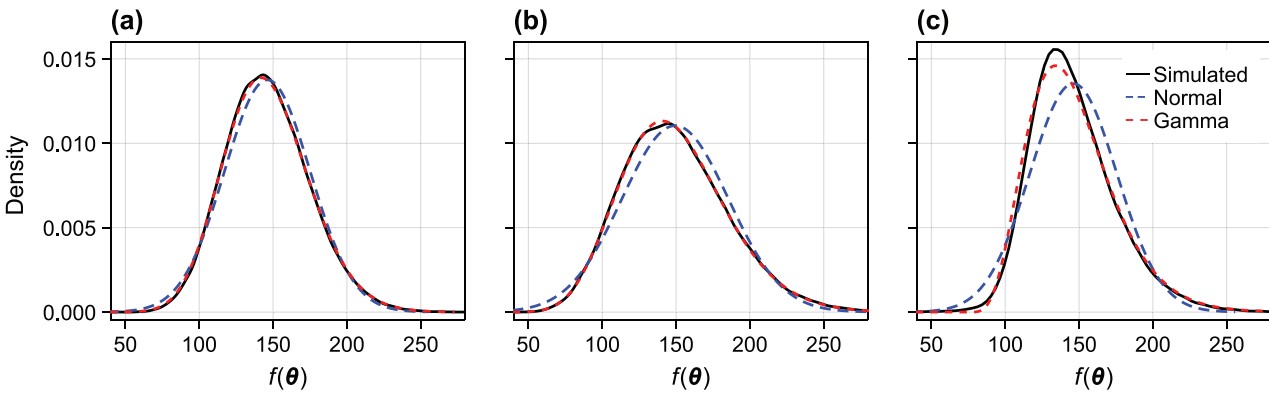

**Fig 2. Accuracy of approximate transformation.** We compare the accuracy of two approximate solutions to the non-linear transformation $f(\theta)$ (here, $f(\theta)$ is given by the solution to the logistic model and $\theta = [\lambda, K, r_0]^\top$, see Eq (24)). In (A) $\theta$ has a multivariate normal distribution with independent components; in (B) $\theta$ has a multivariate normal distribution with correlation between $\lambda$ and $K$; and, in (C) $\theta$ has independent, translated-gamma components such that the marginals have relatively strong skewnesses of $(\omega_\lambda, \omega_K, \omega_{r_0}) = (1, -1, 0.2)$. In all cases, we show a kernel density estimate produced from $10^5$ samples, the normal approximation (blue dashed), and the translated gamma approximation (red dashed). In the supporting material (Table A in S1 File) we provide a statistical comparison between the simulated and approximate distributions using the Kolmogorov-Smirnov test.

that $f(\theta)$ is two-dimensional by correlating $f_1(\theta)$ and $f_2(\theta)$ using a Gaussian copula, a statistical object that correlates two random variables with arbitrary univariate distributions [44]. The correlation parameter in the copula, $\tilde{\rho}$, is chosen to match the approximate correlation calculated from the approximate moments. Denoting the skewnesses of the marginal distributions as $\omega_1$ and $\omega_2$, we compute the map $(\omega_1, \omega_2, \rho) \to \tilde{\rho}$ ahead of time using the rectangle rule for a wide range of skewness parameters $|\omega_i| \in [0, 2]$. The upper limit of 2 is chosen as the gamma distribution changes shape from non-monotonic (normal-like) to monotonic (exponential-like) at $\omega = 2$.

We demonstrate these approximations in Fig 2 using the logistic model (Eq (24)). Additional comparisons, including a multivariate comparison, are provided in the supporting material (Figs A to C in S1 File). In all cases, the gamma approximation is clearly superior to the normal approximation, and provides a fast, accurate approximate approximation to the solution to the random parameter model.

## Surrogate likelihood

We make the assumption that the parameters in the dynamical model, $\theta$, are random variables with a distribution parameterised by $\xi$,

$$\theta \sim D(\xi). \tag{17}$$

For example, $D(\xi)$ may represent a multivariate normal distribution with means, variances and covariances determined by $\xi$ [45]. The only constraint on $D(\xi)$ that we require is that analytical expressions for the moments of $\theta$ be available. For example, we can capture skewness in parameter inputs by describing $\theta$ as having independent components with translated gamma marginals, but cannot, in general, describe the dependence in $\theta$ using a copula. We can capture systems with distinct subpopulations by specifying $D(\xi)$ as a finite mixture, in which case the transformation defined by the mathematical model is applied to each component of the mixture before the mixture is reapplied (in this case, $\xi$ may include parameters relating to both the parameterisation of the mixture components and the mixture weights). Finally, parameters

assumed to be constant can be modelled by assuming they follow a degenerate (i.e., point mass) distribution.

We can form an approximate expression for the likelihood of the data, $\{y_n^{(t_i)}\}_{n=1}^N$ using the approximate solution to the random parameter problem given in Eqs (15) and (16). For a given distribution $\theta \sim D(\xi)$, the moments of $\theta$ will depend on $\xi$; i.e., $\hat{\theta} = \hat{\theta}(\xi)$, $\mathbb{V}(\theta) = m_2(\xi)$, etc, where $m_i(\xi)$ are tensor-valued functions of $\xi$ defined by the specification of $D(\xi)$. Therefore, the moments of $f(\theta)$ can also be expressed as functions of the hyperparameters, $\xi$, such that $\mu = \mu(\xi)$, $\Sigma = \Sigma(\xi)$, and $\omega = \omega(\xi)$. We denote by $p_{f^{(i)}}(y_n^{(t_i)}; \xi)$ the probability density function for $y_n^{(t_i)} \sim f^{(i)}(\theta)$ given $\theta \sim D(\xi)$. For the normal approach, we have that

$$p_{f^{(i)}}(y_n^{(t_i)}; \xi) = \phi(y_n^{(t_i)}; \mu(\xi), \Sigma(\xi)), \tag{18}$$

where $\phi(y; \mu, \Sigma)$ is the density function for the multivariate normal distribution with mean $\mu$ and covariance matrix $\Sigma$. Therefore, the log-likelihood function for snapshot data is given by

$$\ell(\xi) = \sum_{t_i \in \mathcal{T}} \sum_{n=1}^N \log p_{f^{(i)}}(y_n^{(t_i)}; \xi). \tag{19}$$

While not a focus of the present work, the log-likelihood function for time-series data would, therefore, be given by

$$\ell(\xi) = \sum_{n=1}^N \log p_f(y_n; \xi),$$

where $y_n$ includes a set of dependent measurements from all time-points simultaneously, as per Eq (4).

## Inference

As our method provides an approximate likelihood function, we permit application of the full gamut of likelihood-based inference and identifiability techniques. We demonstrate our method using both a frequentist method based on profile likelihood [9], and a Bayesian method based on Markov-chain Monte Carlo (MCMC) [7, 11, 46].

**Profile likelihood.**   We explore identifiability of model hyperparameters using the profile likelihood method [9]. First, we establish the maximum likelihood estimate (MLE) as the hyperparameter vector that maximises the log-likelihood function,

$$\hat{\xi} = \underset{\xi}{\mathrm{argmax}}\ \ell(\xi). \tag{20}$$

Secondly, the hyperparameter space is partitioned, $\xi = [\phi, \lambda]^\intercal$, where $\phi$ is the variable to be *profiled* and $\lambda$ contains the remaining parameters. Profile log-likelihoods, $\hat{\ell}_p$ are then computed for each value of $\phi$ by determining the supremum of the log-likelihood over $\lambda$ relative to the MLE

$$\hat{\ell}_p(\phi) = \sup_{\lambda} \ell(\phi, \lambda) - \ell(\hat{\xi}). \tag{21}$$

We take the supremum of the log-likelihood function using the Nelder-Mead algorithm over a sufficiently large region to cover the true parameters over several orders of magnitude [47].

One interpretation of $\hat{\ell}_p(\phi_0)$ is that of the test statistic in a likelihood-ratio test for $\phi = \phi_0$ [48]. Therefore, approximate 95% confidence intervals for each variable $\phi$ can be constructed

by considering the region where the likelihood-ratio test yields $p$-values greater than $\alpha = 0.05$, corresponding to the region where

$$\hat{\ell}_p(\phi) > \frac{-\Delta_{1,0.95}}{2} \approx -1.92, \qquad (22)$$

where $\Delta_{v,1-\alpha}$ is the $1 - \alpha$ quantile of the $\chi^2(v)$ distribution. Given this interpretation, we can quantify statistical evidence for the presence of variability in a model parameter by examining the profile likelihood in the limit that the variance goes to zero.

**Markov-chain Monte Carlo.**   To obtain samples from the *posterior distribution* of model hyperparameters and hence a distribution that quantifies uncertainty in model predictions, we also perform analysis using a Bayesian MCMC approach [7, 46].

Before consideration of data, $\mathcal{X}$, information about model hyperparameters is encoded in a prior distribution, $p(\xi)$. We then update our knowledge of the parameters using the likelihood to obtain the *posterior distribution*,

$$p(\xi|\mathcal{X}) \propto \exp\left(\ell(\xi; \mathcal{X})\right)p(\xi). \qquad (23)$$

To keep our results consistent with those obtained using the profile likelihood approach, we take the prior distribution to be uniform over the region that covers the true parameters by several orders of magnitude. Therefore, the posterior distribution, Eq (23), is directly proportional to the likelihood function and the MLE corresponds precisely to the maximum *a posteriori* estimate (MAP); we find the MAP by maximising likelihood function using the Nelder-Mead algorithm [47].

We implement an adaptive MCMC algorithm based on the adaptive Metropolis algorithm from the `AdaptiveMCMC` package in Julia [49]. To obtain a posterior predictive distribution of model outputs (in our case, including the probability density function of random parameter distributions) by repeated simulation of the model at posterior samples obtained using MCMC.

## Results

Using the surrogate likelihood based on the moment-matching approximation, we provide a didactic guide to assessing the identifiability of dynamical models with random parameters using three case studies. As our focus is on identifiability, and not model selection, we work using purely synthetically generated data and apply our statistical methodology to recover the true parameter values.

### Logistic model

We first assess identifiability of the canonical logistic model [21]. The logistic model is ubiquitous in biology, ecology, and population dynamics. Our motivation is to describe the time-evolution of the radius of multicellular tumour spheroids (Fig 1) and determine if variability in the initial spheroid size, growth rate, and carrying capacity are identifiable and distinguishable from measurement noise.

Denoting the spheroid radius $r(t)$, the logistic model is

$$\frac{\mathrm{d}r(t)}{\mathrm{d}t} = \frac{\lambda}{3}\, r(t)\left(1 - \frac{r(t)}{R}\right) \text{ subject to } r(0) = r_0, \qquad (24)$$

with exact solution

$$r(t; \boldsymbol{\theta}) = \frac{R}{1 + \left(\dfrac{R}{r_0} - 1\right) \exp\left(-\dfrac{\lambda}{3}t\right)}. \tag{25}$$

Here, $\lambda$ is the per-volume growth rate of the spheroid for $r(t)/R \ll 1$ (the term $\lambda/3$ represents the corresponding per-radius growth rate), $R$ is the carrying capacity radius, and $r_0$ is the initial radius.

We consider data comprising independent measurements (for example, originating from experiments that must be destroyed to collect measurements) subject to additive normal noise such that

$$f^{(i)}(\boldsymbol{\theta}) = r(t_i; \boldsymbol{\theta}) + \varepsilon. \tag{26}$$

Here, $N = 10$ measurements are taken at each $t_i = 2(i - 1)$, $i = 1, 2, \ldots, 8$ (Fig 3A and 3B) and $\varepsilon \sim \mathcal{N}(0, \sigma_\varepsilon^2)$ represents homoscedastic additive normal measurement noise. The logistic model is parameterised by the random parameter vector $\boldsymbol{\theta} = [r_0, \lambda, R, \varepsilon]^\mathsf{T}$. We assess the identifiability for several different parametric forms of the distribution of $\boldsymbol{\theta}$.

**Independent normal random parameters.**   First, we explore identifiability of a model where $\boldsymbol{\theta} \sim D(\boldsymbol{\xi})$ is multivariate normal with independent components, such that

$$r_0 \sim \mathcal{N}\left(\mu_{r_0}, \sigma_{r_0}^2\right), \quad \lambda \sim \mathcal{N}\left(\mu_\lambda, \sigma_\lambda^2\right),$$
$$R \sim \mathcal{N}\left(\mu_R, \sigma_R^2\right), \qquad \varepsilon \sim \mathcal{N}\left(0, \sigma_\varepsilon^2\right). \tag{27}$$

Therefore, $\boldsymbol{\xi} = [\mu_{r_0}, \mu_\lambda, \mu_R, \ln\sigma_{r_0}, \ln\sigma_\lambda, \ln\sigma_R, \ln\sigma_\varepsilon]^\mathsf{T}$, where we infer the natural logarithm of the standard deviations to ensure positivity. We show synthetic data generated from this parameterisation of the random parameter logistic model in Fig 3A and 3B, with identifiability results shown in Fig 3C–3E for $\mu_\lambda = 1$, $\mu_R = 300$, $\mu_{r_0} = 50$, $\sigma_\lambda = 0.05$, $\sigma_R = 20$, $\sigma_{r_0} = 3$, $\sigma_\varepsilon = 4$.

We present profile likelihoods for each parameter in Fig 3C using both the normal and gamma approximations. To aid interpretation, we show the normalised profile likelihood along with the threshold for an approximate 95% confidence interval. Model predictions (mean and 95% prediction interval) produced using the MLE are shown in Fig 3B. As expected from existing analysis of the fixed-parameter logistic model [22], all three location parameters (i.e., the means of $\lambda$, $R$ and $r_0$) are identifiable; this can be seen by profile likelihoods with compact support above the threshold for a 95% confidence interval.

As with the location parameters, we find that the standard deviation of carrying capacity, $\sigma_R$, is also identifiable. We expect this since, for sufficiently large observation times, the solution to the logistic model is simply $r(t) = R$, meaning that experimental observations at these later times are simply observations from the distribution $R \sim \mathcal{N}(\mu_R, \sigma_R^2 + \sigma_\varepsilon^2)$. The most interesting result is that for $\sigma_\lambda$, which is only just identified (within 95% confidence) to a relatively compact region; repeating the exercise with a second set of synthetic data yields a profile likelihood for $\sigma_\lambda$ similar to that for $\sigma_\varepsilon$, suggesting one-sided identifiability, meaning that $\sigma_\lambda$ is indistinguishable from zero (i.e., variability in $\lambda$ cannot be detected). Results for $\sigma_{r_0}$ also suggest at one-sided identifiability. Therefore, only variability in $R$ is distinguishable from measurement noise, although given that results for $\sigma_\lambda$ were borderline identifiable, we expect variability in $\lambda$ to be detectable should it be either larger, or as more data become available.

In Fig 3B we also show results where the fixed-parameter model (i.e., parameters $\lambda$, $R$ and $r_0$ are assumed constant) is calibrated to the data from the random parameter model. This represents the standard approach to parameter inference, where variability in the data is typically

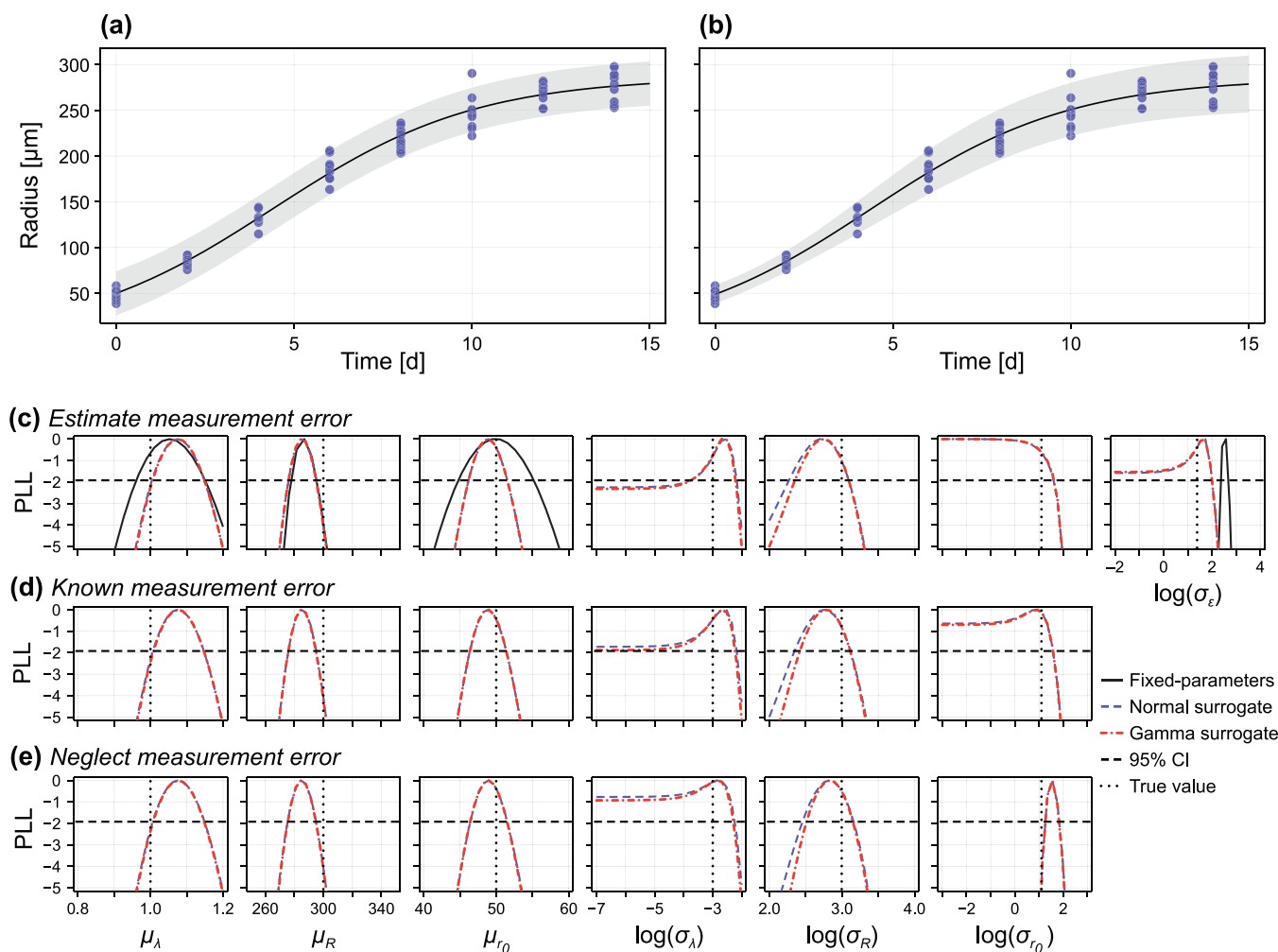

**Fig 3. Profile likelihood analysis for logistic model with random parameters.** We perform inference on a synthetic data set comprising $N = 10$ measurements at $t = 0, 2, 4, \ldots, 14$ of the random parameter logistic model (i.e., 80 independent measurements). In (C) we treat the standard deviation of the measurement noise, $\sigma_\varepsilon$ as unknown, in (D) we assume $\sigma_\varepsilon$ is known (for example, pre-estimated), and in (E) we work with a misspecified model where we assume $\sigma_\varepsilon = 0$, corresponding to a scenario where we assume all variability in the data is due to variability in mechanistic parameters. In (C–E) we compare profiles produced using a normal surrogate model (blue dashed) and gamma surrogate model (red dotted). In (C) we also take a standard inference approach, assuming that observations are normally distributed about model predictions and where parameter variability is neglected (the fixed parameter model). In (A, B) we show the data (blue), model mean (black) and 95% prediction interval at the MLE using (A) the fixed parameter model, and (B) the random parameter approach with a gamma surrogate. Also shown are the true parameter values (black vertical dotted) and a 95% confidence interval threshold (black horizontal dashed).

assumed to comprise entirely of measurement error. Given that the variance of $r_0$ and $\lambda$ were indistinguishable from zero, this may also seem like a reasonable simplification. First, we see that this has relatively little impact on the point estimates for the location parameters, however does give less precise estimates (i.e., wider confidence intervals). As expected, the estimate for $\sigma_\varepsilon$ is larger than the true value, with the true value not contained with the 95% confidence interval; this is expected, since the $\varepsilon$ must now capture both measurement error *and* parameter variability. Examining predictions from the fixed-parameter model in Fig 3A show that accounting for the variation in (at least) carrying capacity produces a much better representation of the variability in the data.

We explore two further scenarios in Fig 3D, where we assumed that measurement error is pre-estimated or known prior to inference, and Fig 3E, where measurement error is neglected (i.e., variability in the data only comes from variability in the parameters). Both cases yield similar results (in terms of point estimates and precision) for the location parameters. Intriguingly, perhaps because $\sigma_\varepsilon$ is relatively small, pre-estimating the measurement error has very little impact on the results for the variance parameters. Finally, neglecting measurement error produces a bias in the estimates for $\sigma_{r_0}$ (which we expect, since $r(0) \sim \mathcal{N}(\mu_{r_0}, \sigma_{r_0}^2 + \sigma_\varepsilon^2)$).

In all cases examined for the logistic model with independent multivariate normal parameters, only very minor differences are observed between results that use the normal and gamma approximations, which we interpret to suggest that the third moment (captured by the gamma approximation, but not the normal approximation) contains very little information about parameter variability.

In the supporting material (Fig F in S1 File), we explore the ability of our method to infer parameter distributions that are not from the exponential family; namely, where the input distributions of the logistic model are independent and uniformly distributed. Despite a discrepancy in higher order moments between the approximate solutions and simulations, we are still able to accurately recover the moments of the input distributions.

**Correlated and skewed random parameters.** Next, we consider two scenarios relating to the complexity of $D(\boldsymbol{\xi})$, the first where $\lambda$ and $R$ are correlated such that

$$(\lambda, R) \sim \text{MvNormal}\left(\begin{bmatrix} \mu_\lambda \\ \mu_R \end{bmatrix}, \begin{bmatrix} \sigma_\lambda^2 & \rho_{\lambda R}\sigma_\lambda\sigma_R \\ \rho_{\lambda R}\sigma_\lambda\sigma_R & \sigma_R^2 \end{bmatrix}\right), \tag{28}$$

and where $r_0$ and $\varepsilon$ are as described by Eq (27). In the second scenario, the growth rate $\lambda$ is skewed such that

$$\lambda \sim \text{ShiftedGamma}(\mu_\lambda, \sigma_\lambda^2, \omega_\lambda), \tag{29}$$

where $r_0$, $R$, and $\varepsilon$ are described by Eq (27) and we set $\rho_{\lambda R} = 0.6$ and $\omega_\lambda = -1.5$.

To assess identifiability of the additional parameter in each of the correlated and skewed models, we show profile likelihoods in Fig 4 for both the normal and gamma approximations, for various sample sizes (observations are taken at the same observation times as in Fig 3). We suppose that prior knowledge has constrained $|\rho_{\lambda R}| < 0.9$ and $-2 < \omega_\lambda < 1$.

Given the lack of identifiability of many parameters in Fig 3, it is anticipated that both additional parameters will be unidentifiable for small sample sizes, as is the case for $N = 10$ observations per time point. Even for a relatively large sample size of $N = 100$ (corresponding to a total of 800 independent samples across all time points), it is only the sign of the skewness parameter that can be identified in the skewed model, whereas the direction of the correlation between $\lambda$ and $R$ cannot be identified to within a 95% confidence interval until a sample of size $N = 1000$ is reached.

Results for the correlated model are similar between the normal and gamma approximations, which we interpret to suggest that higher-order moments in the data do not provide significant additional information about the correlation parameter in the model. In contrast, the results between the normal and gamma approximations are striking; in Fig 4B the normal approximation gives misleading results that suggest that the skewness parameter is non-identifiable even for very large sample sizes.

**Inference for a misspecified random parameter distribution.** Finally, we explore how misspecification of a parameter distribution affects identifiability and model predictions. We consider two cases for the true parameter distribution, the first where $\lambda$ has a strong negative

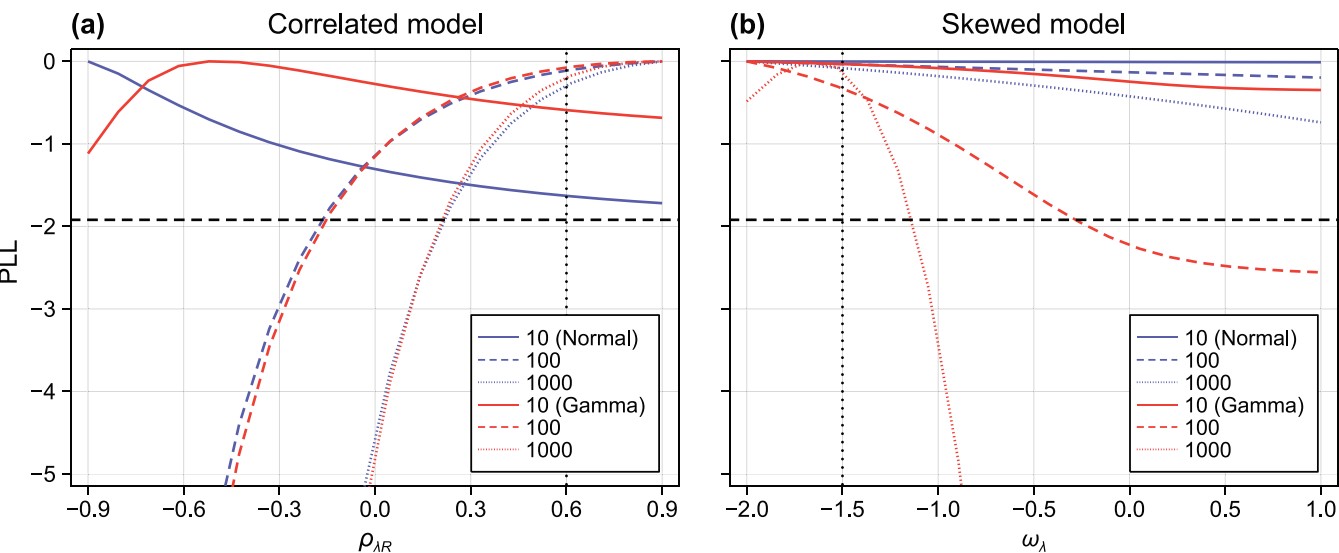

**Fig 4. Profile likelihoods for an unknown correlation coefficient and growth rate distribution skewness.** (A) We infer hyperparameters from synthetic data where the model parameters ($\lambda$, $R$, $r_0$, $\varepsilon$) are multivariate normal as in Fig 3, but with single unknown correlation $Cor(\lambda, R) = \rho_{\lambda R} = 0.6$. (B) We infer hyperparameters from synthetic data where the model parameters are independent as in Fig 3, but where $\lambda$ has a skewed distribution with unknown skewness $\omega_\lambda = -1.5$. Only (A) $\rho_{\lambda R}$ and (B) $\omega_\lambda$ are profiled. In both cases, we produce results using synthetic data sets of size $N = 10$ (solid), $N = 100$ (dashed), and $N = 1000$ (dotted), where normal (blue) and gamma (red) surrogates are used (black horizontal dashed).

skew given by Eq (29), and secondly where $\lambda$ has a bimodal distribution, given by a normal mixture $\lambda \sim w\lambda_1 + (1 - w)\lambda_2$ where

$$
\begin{aligned}
\lambda_1 &\sim \mathcal{N}\left(\mu_\lambda^{(1)}, \sigma_\lambda^{(1)}\right), \\
\lambda_2 &\sim \mathcal{N}\left(\mu_\lambda^{(2)}, \sigma_\lambda^{(2)}\right).
\end{aligned}
\tag{30}
$$

A similar problem was previously explored by Banks et al. [50]. We set $\mu_\lambda^{(1)} = 0.9$, $\mu_\lambda^{(1)} = 1.1$, $\sigma_\lambda^{(1)} = \sigma_\lambda^{(2)} = 0.05$ and $w = 0.4$ (Fig 5E). The bimodal growth rate might represent a situation where, i.e., multiple subpopulations or cell lines are present in the experimental data.

Given that the results in Fig 4 suggest that large sample sizes are required to infer higher-order parameters, such as the skewness of the growth rate, we consider synthetic data generated with $N = 1000$ observations per observation time. We show violin plots of the synthetic data in Fig 5C and 5G for the skewed and bimodal scenarios, respectively. To explore uncertainty in the inferred parameter *distributions* (in contrast to hyperparameter uncertainty), we take a Bayesian approach to inference, and perform inference using MCMC. We perform the analysis using both the true distribution for $\lambda$ (with additional hyperparameters as appropriate), and a misspecified model where $\lambda$ is assumed to be normally distributed.

In Fig 5A and 5E we compare the true distribution with the MAP point estimate and credible intervals for the probability density function for $\lambda$ using posterior samples obtained using MCMC for each model. Given the large sample size, we find that the distribution is identifiable in both cases, confirming our hypothesis from Fig 3 where we found the variance of $\lambda$ to be only one-sided identifiable from a small sample size. In Fig 5B and 5F we show similar results for a misspecified model where $\lambda$ is incorrectly assumed to be normally distributed. These results show that misspecification can sometimes yield over-confidence in the identifiability of parameter density functions: results in Fig 5B and 5F show a narrow 95% credible interval for

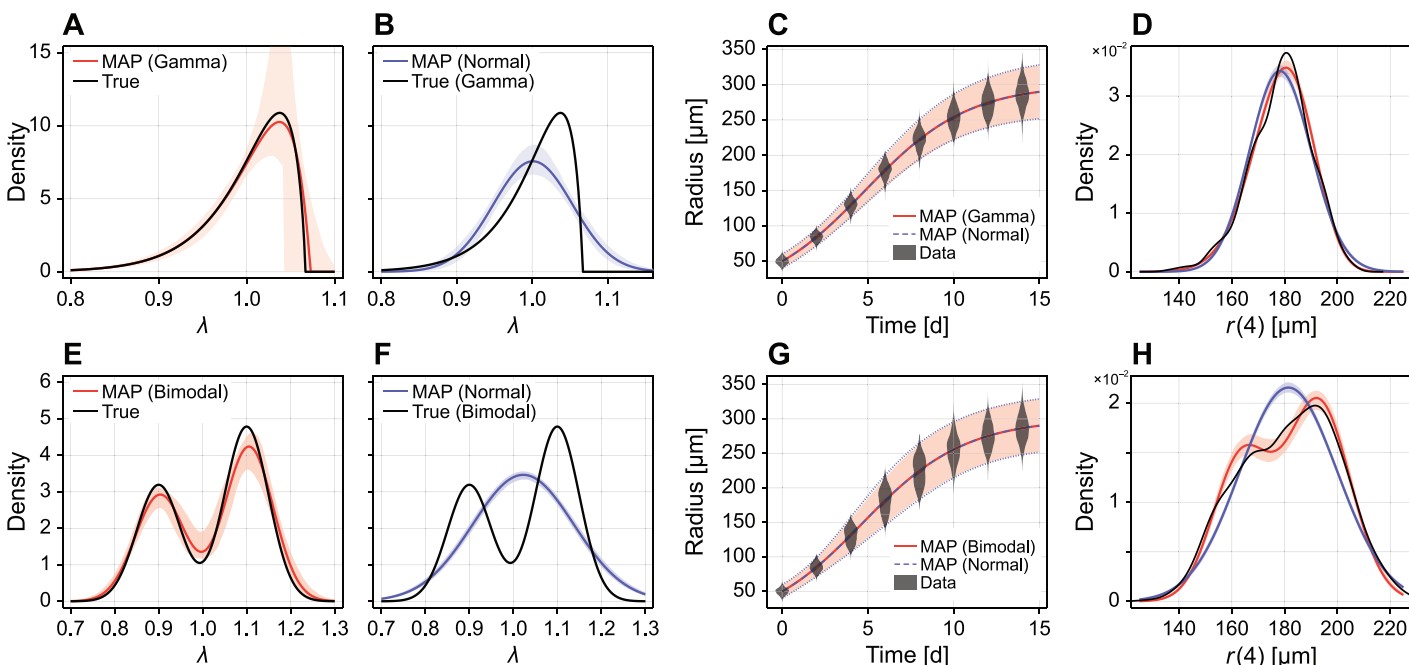

**Fig 5. Inference and prediction where parameter distribution is misspecified.** We explore a case where the underlying growth rate distribution has (A–C) a skewed distribution with $(\mu_\lambda, \sigma_\lambda, \omega_\lambda) = (1, 0.05, -1.5)$, and (e–g) a bimodal distribution, modelled as the mixture $w\lambda_1 + (1-w)\lambda_2$ with $\lambda_1 = \mathcal{N}(0.9, 0.05^2)$, $\lambda_2 = \mathcal{N}(1.1, 0.05^2)$ and $w = 0.4$. To ensure identifiability, we use a large sample size of $N = 1000$ per time point. In (A,E) the true form of the growth rate distribution is used, whereas in (B,F) the growth rate distribution is misspecified and assumed to be normal. Shaded regions in (A,B,E,F) indicate 95% credible intervals for the density. In (C,G), predictions at the MAP estimates (equivalent to MLE) are compared to the data. A 95% prediction interval is shown for the true model (shaded) and the misspecified model (blue dashed), solid curves to the mean, and violin plots show the data. In (D,H), we compare predictions for the density from the true and misspecified models at $t = 4$.

the probability density function for $\lambda$ that do not contain the true distribution. However, it is still possible to accurately infer the statistical moments of the parameter distributions despite misspecification. For example, the true bimodal distribution and inferred MAP normal distribution (Fig 5F) have similar means (1.020 and 1.016, respectively) and variances ($1.22 \times 10^{-2}$ and $1.21 \times 10^{-2}$, respectively).

Results in Fig 5C and 5G, showing a 95% prediction interval for the data at the MAP for both the true and misspecified models, demonstrate that coarse-scale predictions from the misspecified model can be useful. We note, however, that this is not always the case; Banks et al. [50] show that misleading predictions can result when $\lambda$ has a bimodal distribution where $\lambda_1$ and $\lambda_2$ are sufficiently different (in our case, they are relatively similar). In Fig 5D and 5H we show a finer-scale comparison of the predictions from each model by considering a comparison between the predicted probability density function for the tumour spheroid radius at $t = 4$ days, $r(4)$ (MAP with credible intervals). For the skewed model, Fig 5D, predictions are similar between the data (kernel density estimate), true model and misspecified model. However, the misspecified model cannot capture the multimodality of the data at $t = 4$ days, which is captured by the true model (Fig 5). Both the true and misspecified models have similar non-negligible support and (from results in Fig 5G) comparable 2.5% and 97.5% quantiles. In the supporting material (Fig D in S1 File), we demonstrate how the quality of fit obtained from the misspecified model is poor in the case where subpopulations are more distinct $(\mu_\lambda^{(1)} = 0.7, \mu_\lambda^{(2)} = 1.3)$.

## Linear two-pool model

The transfer of chemical species between and from two-pools is used widely as a model of cholesterol transfer or urea decay [51, 52]. We consider that material transfers from species one, denoted $X_1$, to species two, denoted $X_2$, at rate $k_{21}$ and decays from each pool at rates $k_1$ and $k_2$, respectively. That is, we consider the chemical model

$$X_1 \xrightarrow{k_{21}} X_2 \xrightarrow{k_2} \emptyset,$$
$$X_1 \xrightarrow{k_1} \emptyset, \tag{31}$$

which we describe using a coupled system of differential equations describing the time-rate of change of the concentration of material in each pool, $x_1(t)$ and $x_2(t)$, respectively, so that

$$\frac{\mathrm{d}x_1(t)}{\mathrm{d}t} = -(k_{21} + k_1)x_1(t),$$
$$\frac{\mathrm{d}x_2(t)}{\mathrm{d}t} = k_{21}x_1(t) - k_2x_2(t). \tag{32}$$

We model a closed system subject to a known input at $t = 0$ such that $x_1(0) = x_0$ and $x_2(0) = 0$.

We consider an inference problem where observations are taken from only the second pool and that the measurement error scales with the concentration. Therefore, we assume multiplicative normal noise, such that

$$f^{(t_i)}(\boldsymbol{\theta}) = x_2(t_i)\varepsilon. \tag{33}$$

We further assume that the decay from each pool is due to a strictly chemical process such that $k_1$ and $k_2$ are constant, and that $k_{21}$ is a normally distributed random variable. Variation in $k_{21}$ between data might arise clinically from variability between patients. We incorporate this parameterisation into our framework by assuming that $\boldsymbol{\theta} = [k_1, k_{21}, k_2, \varepsilon]^{\mathsf{T}}$ is a random parameter vector with independent components, where

$$k_1 \sim \delta(\mu_1), \qquad k_2 \sim \delta(\mu_2),$$
$$k_{21} \sim \mathcal{N}\big(\mu_{21}, \sigma_{21}^2\big), \quad \varepsilon \sim \mathcal{N}(1, \sigma^2). \tag{34}$$

Here, $\delta(x)$ denotes a Dirac or degenerate distribution (we take $\delta(x)$ to be normally distributed with $\sigma \to 0$ such that all central moments above the third are zero). For the linear two-pool model, we have that $\boldsymbol{\theta} \sim D(\boldsymbol{\xi})$ with hyperparameters $\boldsymbol{\xi} = [\mu_1, \mu_{21}, \mu_2, \ln \sigma_{21}, \ln \sigma]^{\mathsf{T}}$. We set $\mu_1 = 0.7$, $\mu_{21} = 0.6$, $\mu_2 = 0.4$, $\sigma_{21} = 0.1$ and $\sigma = 0.01$, and generate synthetic data using $N = 20$ independent observations at $t \in \{0.5, 1.5, 2.5, 3.5, 5, 7\}$ (Fig 6G). The solution to Eq (31) at the MLE is shown in Fig 6G, and the approximate solution based on both a normal and gamma approximation is given as supporting material (Fig B in S1 File) in addition to a statistical comparison (Table B in S1 File). Given that the distribution of material in the second pool is skewed at later times, we work only with the gamma approximation for analysis of the two-pool model.

Profile likelihood results in Fig 6A–6E show that all physical parameters are identifiable, including the variance in the transfer rate $k_{21}$. This result is particularly interesting as we are able to identify the source of variance due to heterogeneity despite the variance of the measurement noise being only one-sided identifiable; we cannot distinguish a model with measurement noise from a model without measurement noise, but this has no impact on the identifiability of other model parameters. Results in Fig 6G show model predictions (mean and 95% prediction interval) computed using the MLE. Evident in Fig 6G is a key advantage of the random parameter approach—in contrast to the standard approach where variability is

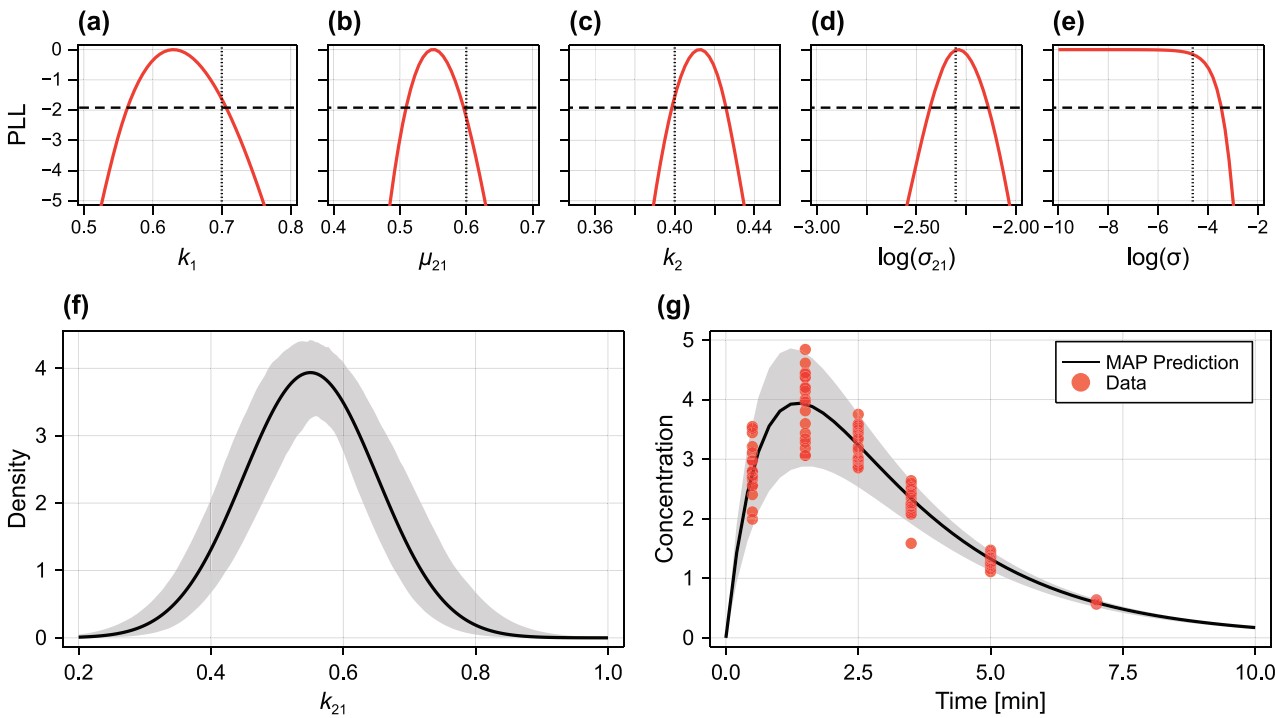

**Fig 6. Identifiability analysis for two-pool model with random parameters.** (A–E) Profile likelihoods for each hyperparameter. Also shown are the true values (vertical dotted) and the threshold for a 95% confidence interval (horizontal dashed). (F) Inferred distribution of $k_{21}$ showing the distribution at the MAP (black) and a 95% credible interval for the density function. (G) Synthetic data (red discs) and model prediction based on the MLE showing the mean (black) and 95% prediction interval (grey).

often assumed to originate from independent measurement noise—where our model produces not only average behaviour consistent with the data, but excellent predictions relating to the data variance.

To better visualise how well the unknown distribution of $k_{21}$ is identified from the available data we repeat the identifiability analysis taking a Bayesian approach to obtain posterior samples using MCMC. In Fig 6F we show a predictive distribution of the density function for $k_{21}$ (mean and 95% credible interval of the density function), showing that the distribution is identifiable and that relatively precise estimates are recovered.

## Non-linear two-pool model

Finally, we consider a non-linear extension of the two-pool model where the transfer rate is not constant, but described by a non-linear Michaelis-Menten form, $k(x_1) = k_{21}x_1/(V_{21} + x_1)$. That is, we consider the chemical model

$$X_1 \xrightarrow{k(x_1)} X_2 \xrightarrow{k_2} \emptyset,$$
$$X_1 \xrightarrow{k_1} \emptyset, \tag{35}$$

described by the system of differential equations

$$\frac{dx_1(t)}{dt} = -\left(\frac{k_{21}}{V_{21} + x_1(t)} + k_1\right)x_1(t),$$
$$\frac{dx_2(t)}{dt} = \frac{k_{21}x_1(t)}{V_{21} + x_1(t)} - k_2x_2(t). \tag{36}$$

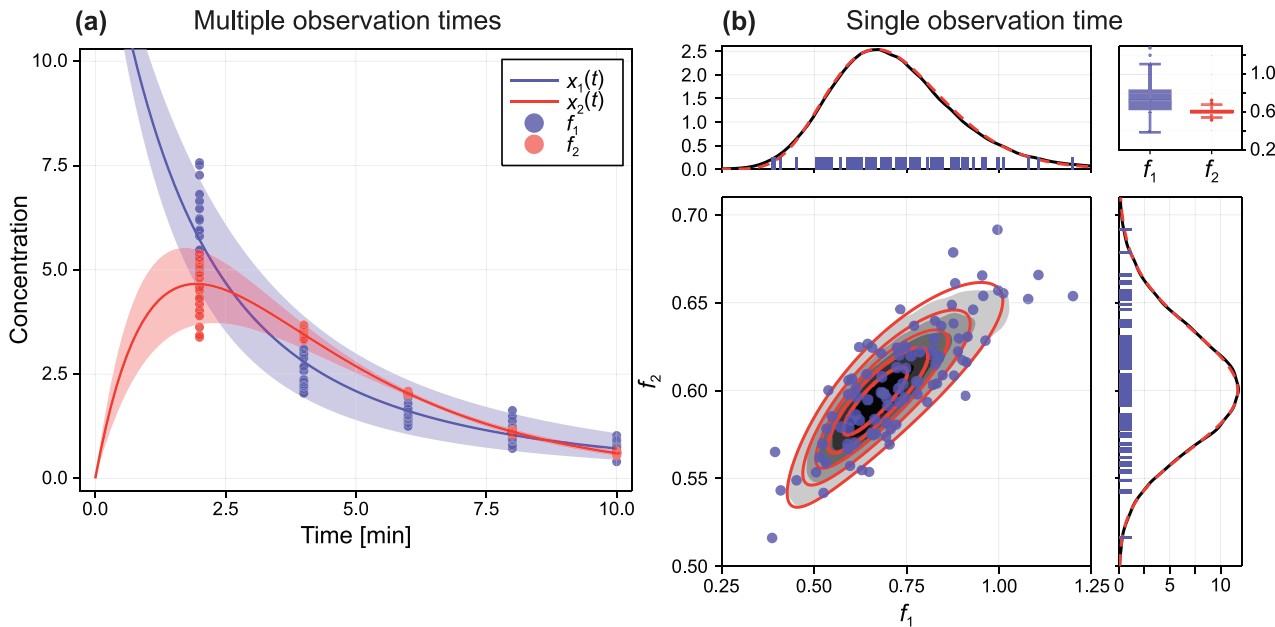

**Fig 7. Synthetic data from the non-linear two-pool model.** (A) Data comprise $N = 20$ noisy observations of the concentration in each pool from five observation times. Also shown is the mean and a 95% prediction interval based on the approximate solution to the random parameter non-linear two-pool model. Bivariate data and solution to the random parameter problem are provided as supporting material (Fig C in S1 File) in addition to a statistical comparison for each marginal distribution (Table C in S1 File). (B) Data comprise $N = 100$ noisy observations from the single observation time $t = 10$ (blue discs). Also shown is the approximate solution to the random parameter problem using correlated gamma marginals (red solid), and the exact density based on $10^5$ randomly sampled parameter values (grey filled).

In contrast to the previous two case studies, an exact solution is not available for the non-linear two-pool model. Therefore, this case study provides an example of the flexibility of our approach: we can solve the non-linear two-pool model using an explicit numerical scheme [53] and use automatic differentiation [54] to calculate the necessary derivatives with minimal additional computational overhead.

We consider identifiability under two scenarios. In both cases, we collect bivariate (i.e., dependent) outputs from both pools, with measurements of pool one subject to multiplicative normal noise, and that of pool two subject to additive normal noise, such that

$$f^{(t_i)}(\boldsymbol{\theta}) = \begin{bmatrix} x_1(t_i)\varepsilon_1 \\ x_2(t_i) + \varepsilon_2 \end{bmatrix}. \tag{37}$$

In the first scenario, we take $N = 20$ bivariate observations at several observation times; $t_i = 2i$ for $i = 1, 2, \ldots, 5$ (observations at different observation times are independent). Synthetic data for the first scenario are shown in Fig 7A (bivariate data with the gamma approximation are shown in supporting material (Fig C in S1 File)). In the second, clinically and experimentally motivated scenario [55, 56], we consider that $N = 100$ observations are available from the single observation time $t = 10$. This second scenario represents a situation where, for example, the data collection method is invasive or possibly where patients must return to a clinic for data collection. Synthetic data for the second scenario are shown in Fig 7B. Given that univariate observations are skewed, we consider only the bivariate gamma approximation for analysis of the non-linear two-pool model. Results in Fig 7B show excellent agreement between the synthetic data and gamma approximation.

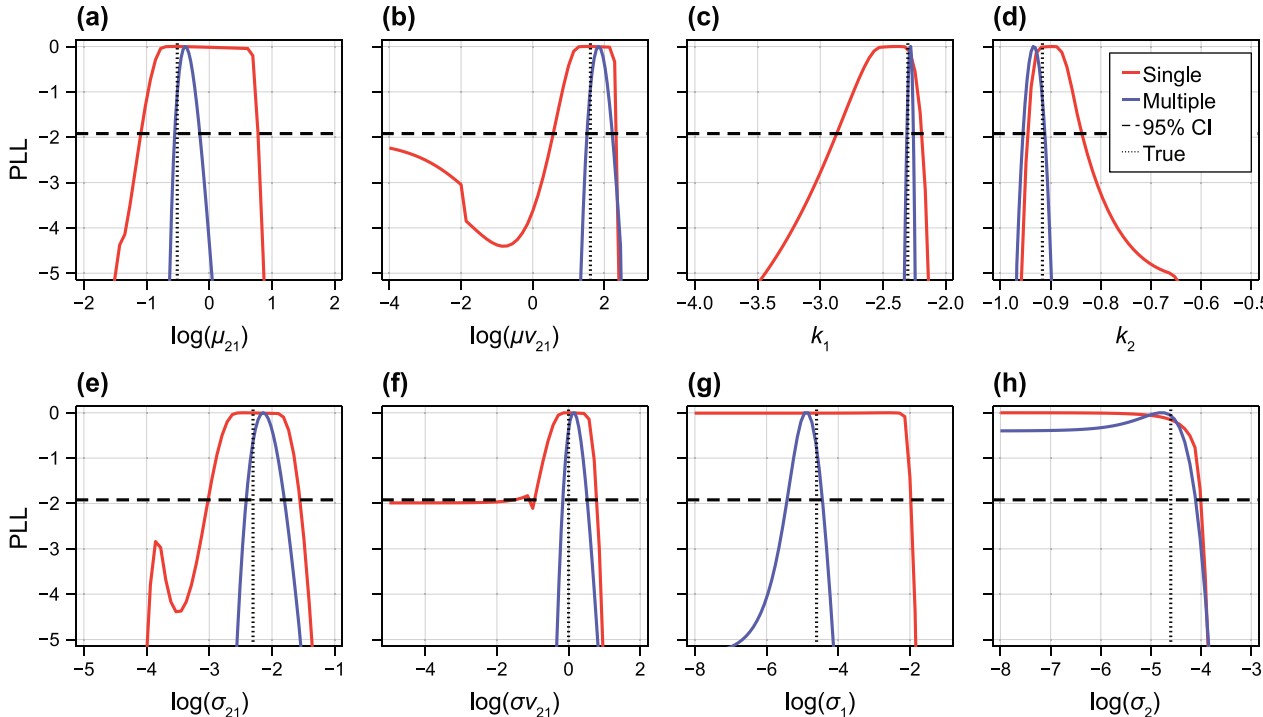

**Fig 8. Identifiability analysis for the non-linear two-pool model.** Profile likelihoods for each parameter where the data comprise $N = 20$ observations each from five observation times (blue) and $N = 100$ observations at a single observation time.

The random parameter vector $\boldsymbol{\theta} = [k_1, k_{21}, V_{21}, k_2, \varepsilon_1, \varepsilon_2]^\top$ has independent components, where

$$
\begin{aligned}
k_1 &\sim \delta(\mu_1), & k_2 &\sim \delta(\mu_2), & K_{21} &\sim \mathcal{N}\left(\mu_{21}, \sigma_{21}^2\right), \\
V_{21} &\sim \mathcal{N}\left(\mu_{V_{21}}, \sigma_{V_{21}}^2\right), & \varepsilon_1 &\sim \mathcal{N}\left(1, \sigma_1^2\right), & \varepsilon_2 &\sim \mathcal{N}\left(0, \sigma_2^2\right).
\end{aligned}
\tag{38}
$$

We set $\mu_{21} = 0.6$, $\mu_{V_{21}} = 5$, $\mu_1 = 0.1$, $\mu_2 = 0.4$, $\sigma_{21} = 0.1$, $\sigma_{V_{21}} = 1$, $\sigma_1 = \sigma_2 = 0.01$.

Profile likelihood results in Fig 8H show that all parameters are identifiable from data with multiple observation times, with the exception of the standard deviation of the additive normal noise process for pool 2; $\sigma_2$ is one-sided identifiable (indistinguishable from zero). Despite the sample sizes being equivalent, parameter estimates are more precise from data with multiple observation times than from a single observation time.

Results from data with a single observation time are more interesting. At first, it appears that all hyperparameters relating to the physical parameters are identifiable (the variance of $V_{21}$ is border-line identifiable, and the variances of the measurement noise variables are one-sided identifiable). However, the profile likelihood is relatively flat around the MLE. Given that we have taken a moment-matching approach to inference, we explore this further by exploring the sensitivity matrix, or Fisher information matrix (FIM) of the function $M(\boldsymbol{\xi}) : \mathbb{R}^8 \to \mathbb{R}^7$, which maps the hyperparameters to the moments of the output. The FIM is given by

$$
\boldsymbol{S}(\xi) = \boldsymbol{J}_M(\xi)^\top \boldsymbol{J}_M(\xi)
\tag{39}
$$

where $J_M(\boldsymbol{\xi})$ is the Jacobian of $M(\boldsymbol{\xi})$. The FIM relates directly to the Hessian (i.e., curvature) of the log-likelihood under the assumption that observations of the moments are normally distributed. Furthermore, the rank of the FIM at the MLE gives insight into the local-identifiability of the model: for identifiability, we require that FIM be of full-rank (or equivalently, non-singular) [43]. Using automatic differentiation to find $J_M(\hat{\boldsymbol{\xi}})$ we find that $S(\hat{\boldsymbol{\xi}})$ has one zero eigenvalue so that $\mathrm{rank}(S(\hat{\boldsymbol{\xi}})) = 7 < 8$. Therefore, parameters are locally non-identifiable; we also see this from profile likelihood analysis in Fig 8. We have not in this work explored the connection between the identifiability of the fixed parameter model and that of the random parameter model. This question is particularly relevant for the single observation time example as we would not, in general, expect that the four biophysical parameters $[k_1, k_{21}, V_{21}, k_2]^\top$ be identifiable from a single two-dimensional output. The provision of deterministic expressions connecting the input and output moments (Eqs (11) to (14)) may allow more rigorous exploration of this question in future work.

From profile likelihood results in Fig 8 we also notice non-monotonic behaviour in the likelihood, particularly in Fig 8B and 8E. To explore this further, we take a Bayesian approach to identifiability analysis [7, 11] and explore the convergence of 12 independent MCMC chains of length 100,000, 11 initiated at randomly sampled regions of the prior, and one chain initiated at the true values. In Fig 9A we see that several chains converge to a region of the parameter space with relatively low likelihood, whereas several converge to a region with comparable log posterior density to the MAP. In Fig 9C–9J we explore the marginal density of chains that converge to a region where the mean log-posterior density from the final 60,000 iterations is within a 95% confidence level of the MAP. First, it is clear that results from the single chain initiated at the true value are different from the other chains: the likelihood is clearly multimodal, where regions of the parameter space where the mean or the variance of $V_{21}$ is zero. We demonstrate this in Fig 9B by finding a second MAP for a model where $\sigma_{V_{21}} = 0$, showing that both models are indistinguishable.

## Discussion

Deterministic differential equation models are routinely applied to analyse data in terms of parameters that carry physically meaningful interpretations. Traditionally, these models have fixed parameters that describe only the mean of experimental observations: variability in data is neglected, often assumed to originate from a noise process unrelated to the underlying dynamics (i.e., measurement error) [7, 9, 13]. Allowing model parameters to vary randomly according to probability distributions provides flexibility to account for the heterogeneity that plays an essential role in the emergent behaviour of many biological and biophysical systems. Methods for performing inference of these models, and consequentially assessing parameter identifiability, are traditionally limited by a computational cost that far exceeds that of the corresponding fixed-parameter model. In this work, we present a novel framework for identifiability analysis of differential equation models with random parameters with a computational cost comparable to that of the fixed parameter problem. Our approach is applicable to many existing workflows, since we use a standard class of deterministic, differential equation models, and provide an approximate expression for the likelihood function providing flexibility in terms of the statistical methods used for inference and identifiability analysis.

We approach the random parameter inference problem by specifying a distribution for model parameters, and infer *hyperparameters* that relate to each model parameter distribution. Notably, this approach increases the number of unknown parameters that have to be estimated from data, however also allows interpretation of additional information that may be available in higher-order statistical moments of the data. Identifiability analysis of the logistic model, for

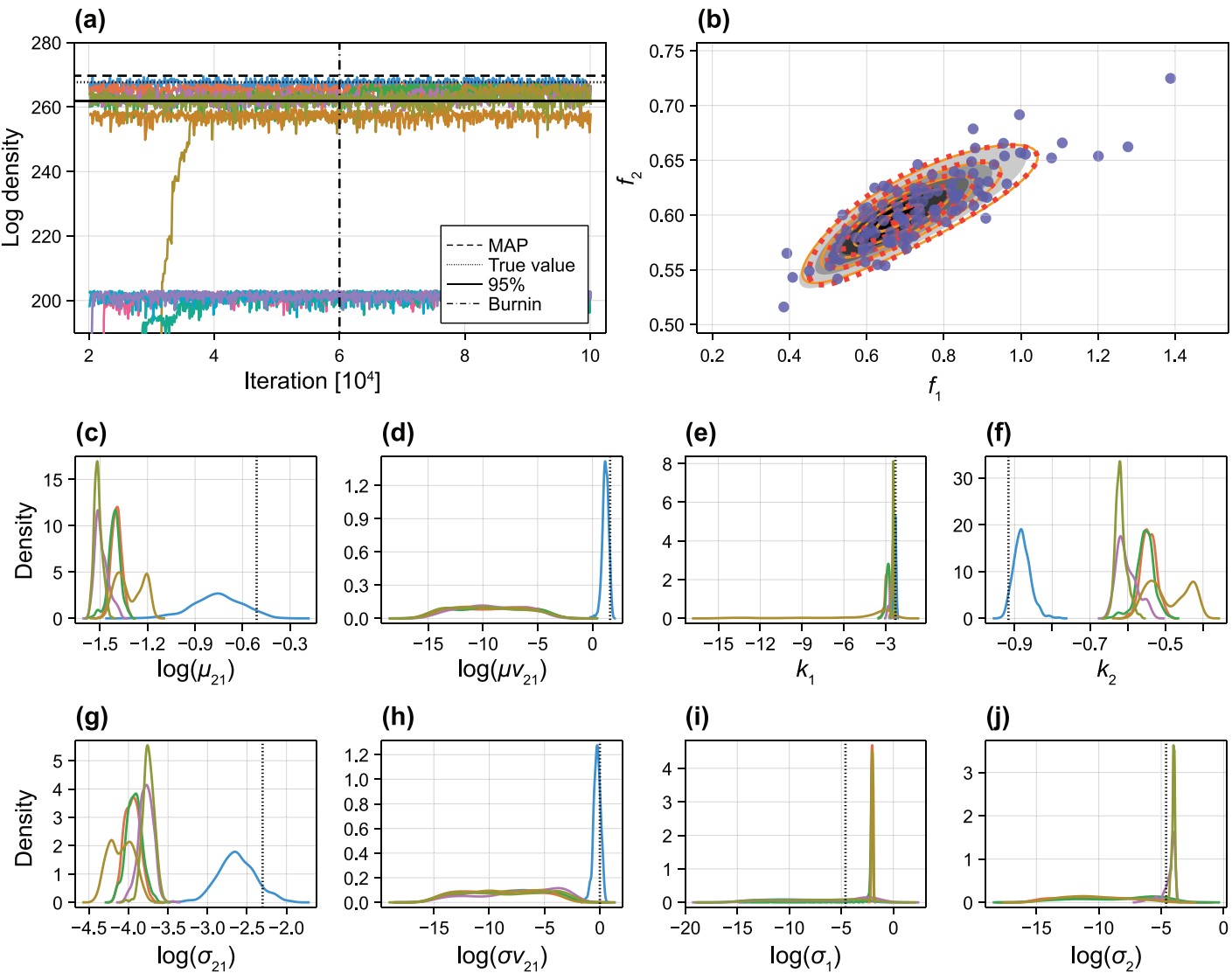

**Fig 9. MCMC results for non-linear two-pool model.** (A) MCMC was run for $10^5$ iterations at twelve initial locations: eleven sampled from the prior and one at the true value. Coloured curves show convergence in the log posterior density. Also shown is the posterior density at the MAP (dashed), posterior density at the true value (dotted), a 95% threshold based on an asymptotic chi-squared distribution and the MAP (solid). The first $6 \times 10^4$ iterations were discarded as burn-in. Each colour corresponds to an independent MCMC chain. (B) Synthetic data (discs), approximate model solution at the MAP (orange solid), approximate model solution at a model where $V_{21}$ has zero variance (red dotted), and approximate model solution at the true values (grey). (C–F) Marginal posterior densities for each parameter. Each colour corresponds to an independent MCMC chain.

example, shows that the additional unknown parameters in the random parameter model (i.e., hyperparameters relating to the variance of each model parameter) do not yield greater uncertainty in the mean of the parameters as is often assumed when the number of unknown parameters in a model increases. In fact, we see in Fig 3C that applying the random parameter model yields more precise estimates of the average proliferation rate and initial spheroid radius. We attribute this, in part, to a more accurate specification of the observation variance: for the fixed parameter model, we assume that variability arises due to homoscedastic normal measurement error, which leads to both under- and over- dispersion at early and late times, respectively (Fig 3C). This can be avoided by allowing the variance to vary with time (for

example, by specifying a functional form for the variance), or accounted for naturally using the random parameter model. We find that our model yields accurate predictions of the data variance despite non-identifiability of several hyperparameters which relate to the model parameter variances.

The computational cost and ease of implementation of our approach is comparable to the fixed parameter model, in contrast to approximate Bayesian computational methods [32, 33], which are computationally costly, and Bayesian hierarchical approaches [27, 28], which suffer from a parameter dimensionality that scales with sample size. We benchmark our approach using the non-linear two-pool model with a single observation time, finding that likelihood evaluations for the random parameter problem (850 μs) are comparable to timings for the fixed parameter problem (67 μs) once inefficiencies in our implementation are considered (for example, forming the four-dimensional kurtosis tensor $\mathbb{K}(\boldsymbol{\theta})$ without exploiting significant sparsity accounts for 65% (550 μs) of the computation time). Computations were performed on an Apple M1 Pro chip. Overall, the second order Taylor series provides an adequate approximation to the models we consider, requiring evaluation of only the model mean, gradient, and Hessian: all of which can be obtained efficiently and with relative ease using automatic differentiation. While the two-moment normal approximation can yield similar results in cases where the data are not significantly skewed, the three-moment approximation provides better results for a wider range of models with only minor additional computational cost. The use of automatic differentiation [54] means that the code we provide for analysis is applicable to a broad class of potentially black-box deterministic models, with any measurement noise model, provided that model outputs are vector valued.

The primary limitation of our approach is that data must be adequately approximated with a normal or gamma distribution, or be expressible as a mixture of normal or gamma distributions. While this may seem restrictive, we note that it is often the case in the mathematical biology literature that data are assumed normally distributed about model predictions, which describe the data mean (or equivalently, models are calibrated using least-squares estimation) [5, 7, 22, 57]. This assumption can be assessed by examining the fit of the approximate distribution to the data at the MLE. In the supporting material (Fig E in S1 File), we demonstrate a pathological example where our model performs poorly, by approximating the solution to the logistic model with a strong Allee effect [58]. The distribution of the initial condition is chosen so that approximately 16% of model realisations lead to population extinction, whereas 84% lead to logistic growth to carrying capacity. The resultant distribution is bimodal and constrained to a finite interval, whereas the approximation is unimodal, has infinite support, and clearly cannot capture the data. As our approximations are constructed from a finite set of moments, our approach may also fail for high-dimensional data where the dependence structure may be highly non-linear and not adequately captured by a multivariate normal distribution; this is potentially the case with time-series data.

Two sources of variability that we do not consider include intrinsic variability arising, for example, from the chemical master equation, and uncertainty in the independent variable. The former can be captured in a differential equation framework through stochastic differential equations [2, 59], potentially allowing for our approximate approach to inference and identifiability analysis through a nested moment-matching approach [12, 60] that captures both intrinsic variability and variability in model parameters. The latter source of variability is clinically relevant; immunological data arising from study of COVID-19 and other infectious diseases [56], relates to highly heterogeneous biological processes, and the exact time of infection is typically unknown. By making a distributional assumption for the infection time, $t$, we can already apply our framework to calculate the conditional distribution of measurements $p(\boldsymbol{x}|t, \boldsymbol{\theta})$. This time-dependent distribution can be constructed efficiently by assuming continuity

and constructing an interpolation of the moments $\boldsymbol{x}$ over a range of measurement times, $t$. The joint distribution of measurements and observation time can then be analytically expressed

$$p(\boldsymbol{x}, t | \boldsymbol{\theta}) = p(\boldsymbol{x} | t, \boldsymbol{\theta}) p(t | \boldsymbol{\theta}), \tag{40}$$

and a likelihood constructed that accounts for uncertain observation times, that are possibly dependent on $\boldsymbol{\theta}$.

Heterogeneity is ubiquitous to biology, playing an essential role in the behaviour of biological systems, and contributing to the variability present in biological data. In this work, we present a novel, computationally efficient, framework for inference and identifiability analysis for differential equation based models that incorporate heterogeneity through random parameters. We demonstrate how our framework can be applied to identify sources of biological variability from data, and produce both more precise parameter estimates and more accurate predictions with minimal additional computational cost compared to a fixed-parameter approach. Our framework is easy to implement and applicable to a wide range of models commonly employed throughout biology. A better understanding of heterogeneity in biology, aided by quantitative methods to extract heterogeneity from data, has potential to yield a better understanding of disease, more accurate predictions and an overall more holistic insight into biological behaviour.

## Supporting information

**S1 File. Supporting material document.**
(PDF)

## Acknowledgments

We thank Nikolas Haass and Gency Gunasingh for training A.P.B. to perform the tumour spheroid experiments that motivated this work.

## Author Contributions

**Conceptualization:** Alexander P. Browning, Christopher Drovandi, Ian W. Turner, Adrianne L. Jenner, Matthew J. Simpson.

**Investigation:** Alexander P. Browning, Christopher Drovandi, Ian W. Turner, Adrianne L. Jenner, Matthew J. Simpson.

**Methodology:** Alexander P. Browning, Christopher Drovandi, Ian W. Turner, Adrianne L. Jenner, Matthew J. Simpson.

**Software:** Alexander P. Browning.

**Supervision:** Matthew J. Simpson.

**Visualization:** Alexander P. Browning.

**Writing – original draft:** Alexander P. Browning.

**Writing – review & editing:** Alexander P. Browning, Christopher Drovandi, Ian W. Turner, Adrianne L. Jenner, Matthew J. Simpson.

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
