## [Decision Letter · Decision Letter 0]

10 Oct 2022

Dear Professor Simpson,

Thank you very much for submitting your manuscript "Efficient inference and identifiability analysis for differential equation models with random parameters" for consideration at PLOS Computational Biology. As with all papers reviewed by the journal, your manuscript was reviewed by members of the editorial board and by several independent reviewers. The reviewers appreciated the attention to an important topic. Based on the reviews, we are likely to accept this manuscript for publication, providing that you modify the manuscript according to the review recommendations. Notice, however, that the reviewers made important comments on some choices in your analysis and how sensitive your results are to them. Please make sure that your revised submission addresses all these concerns.

Sincerely,

Ricardo Martinez-Garcia

Academic Editor

PLOS Computational Biology

Jason Papin

Editor-in-Chief

PLOS Computational Biology

Reviewer's Responses to Questions

**Comments to the Authors:**

Reviewer #1: Browning et al. present a new approach that enables parameter estimation and identifiability analysis for differential equation models incorporating heterogeneity. This approach allows for efficient inference and identifiability analysis by leveraging an approximate solution to the differential equation. Overall this work is widely applicable to differential equation models that explicitly model heterogeneity observed in the data via random parameters with parametric distributions. The efficient approach presented in this work enables critical analyses that would otherwise be computationally intractable for many applications.

Overall, I found that the authors clearly explain their method and provide adequate background and detail to enable a reader’s understanding. Additionally, the authors use three test cases to provide an excellent introduction to the identifiability analysis of this class of models. I believe this work will allow future users to conduct a similar analysis of their models.

I have no major changes to suggest, but I have a few minor points I would like the authors to address:

The proposed approach requires the user to specify parametric distributions for the model parameters. They show that misspecification of these distributions can lead to errors in the identifiability analysis. I am concerned about the limitations of this approach when prior knowledge of these distributions is limited. Can the authors address the possible limitations of assuming specific parametric distributions for model parameters and provide suggestions for when the distribution may be unknown?

How does the identifiability of a model parameter affect the identifiability of its distributions hyper-parameters? For instance, if one found that a parameter is non-identifiable via an a priori analysis, what should they expect to see in the identifiability analysis of that parameter’s hyper-parameters?

Does the identifiability of a parameter’s variance terms provide insight into the model's sensitivity to those parameters? For instance, in section 3.1.1, the authors state that the slight identifiability of the variance parameter for lambda indicates that variability in this lambda cannot be distinguished from measurement noise. Does this imply that the model is not particularly sensitive to lambda’s value?

Can the authors clarify how they determine if a parameter is identifiable from its 95% credible interval in the cases where they perform a Bayesian analysis? For example, in section 3.1.3, lines 393-395, the authors conclude that the distribution of parameter lambda is identifiable. However, it is unclear how they arrived at this conclusion from the results presented in figure 5.

The authors present their method with examples that have relatively few state variables and parameters. Can the authors comment on how this approach scales to models with greater numbers of parameters?

In addition to these points, I found several typos while reading the manuscript. Can the authors please correct the following typos and double-check the manuscript and supplemental materials:

The caption for figure 4 states “omega_lambda = -1.5” however, the corresponding text on line 357 states “omega_lambda = 1.5.” I believe the value on line 357 needs to be negative.

Line 437 says, “second tool.” I believe this should read “second pool.”

Reviewer #2: In their manuscript titled "Efficient inference and identifiability analysis for differential equation models with random parameters", Browning and colleagues introduce a new method for the calibtration of ODE models with random parameters. The model can be used for the description and inference of inter-individual heterogeneity, which is a very relevant problem in the current literature. The proposed method is novel in so far as that noise model is incorporated into the model-transformed random variable, and that the taylor approximation is applied differently than in similar methods such as the method of moments or van kampens system size expansion, which brings some advantages in terms of scalability (potentially at the cost of some accuracy). The paper is very well written and easy to follow, but there are some technical aspects that remain opaque (which I will go into more detail below). I generally like the approach, and definitely think that this paper should be published. However, I am some concerns that I describe below, but I have no doubt the authors will be able to adress them.

Major Points:

1) Embedding in the existing literature:

The paper is a bit heavy on statistical jargon which might make it difficult for readers with a stronger biological background to follow the paper. Specifically it would be great to give a bit more explanation about what the authors mean by random/fixed effects models. I am familiar with these terms in the context of Non-linear mixed effects models, which are likely to be what the authors call "hierarchical" models. Such models have been used in the context of biological models (see e.g., https://doi.org/10.1186/s12918-015-0203-x, https://doi.org/10.1371/journal.pcbi.1004706, https://doi.org/10.1371/journal.pone.0124050, https://doi.org/10.1038/s41540-018-0079-7 and references therein). Similarly the authors should contrast their approach to similar approaches such as https://doi.org/10.1016/j.cels.2018.12.007 or https://doi.org/10.1038/nmeth.2794.

Another range of approaches that seems to be relevant, but isn't really discussed are the moment closure approximations (https://doi.org/10.1063/1.3454685) or van Kampens system size expansion (https://doi.org/10.1063/1.3454685). Both use taylor expansion to approximate moments, but with respect to different variables, which would be helpful if mentioned in the paper. Usually these methods are employed for the description of stochastic models, but, using the approach described in https://doi.org/10.1016/j.cels.2018.04.008 which the authors also use, can also be applied to describe heterogeneity. Accordingly, the authors also should contrast their approach to (https://doi.org/10.1371/journal.pcbi.1005030) where moment closure and van Kampens approximation are used for paramete inference (in a stochastic modeling context, not a heterogeneity context, but the transfer is trivial.).

2) Snapshot vs Timecourse data:

The authors just briefly mention the issue of considering snapshot vs timecourse data (it would be good to mention these terms such that a more biological audience can also follo) in l142-145. However, this isn't really picked up in the remainder of the manuscript, but is quite relevant in practical terms. The key difference for timecourse data is that there is temporal correlation between simulations across timepoints. It is unclear to me how this is accounted for in the method that the authors present, as what the authors describe only looks like an rearrangement of indices.

3) Likelihood function:

I am still unsure whether I am fully grasping what the authors are actually doing. For me it would be quite helpful to have some visual depcition of how the approximation method that the authors are proposing actually works. Similarly, it would probably to explicitely write down the equation for the likelihood pf(y,xi). My understanding is that this would simply, in the case of the normal approach, be a multivariate normal probability density function with mean and standard deviation according to equations 10 and 11. I understand that the equations would be quite bulky, but (10) - (13) already pose a solid chance of scare the reader away ;).

4) Approximation Error:

The authors really only discuss the approximation error in figure 2 and the discussion and seem to forget about in the remainder of the results. It would be good to see some more investigation of the approximation error of the method in the more complex settings, to make sure that the findings are not the result of approximation errors.

Minor Points:

a) I am not sure what the argument about the inverse of f in line 149 is about. In the vast majority of cases f(theta) wont be available analytically because f itself is not available analytically in the first place.

b) l234 should probably read "log-likelihood function" instead of "likelihood function"

c) panel labels in the legend to figure 5 seem misarranged

d) for the parameterization of the covariance matrix D(\\xi) the authors may want to consult https://doi.org/10.1016/j.celrep.2021.109507 for some pointers on how to avoid overparameterization

e) non-identifiability and slopiness are not the same, see https://doi.org/10.1016/j.mbs.2016.10.009

f) it's a bit uncommen to introduce new data in the discussion, the authors may want to create a separate section discussing benchmarking and the model with a strong Allee effect

Reviewer #3: Browning et al present methods for the analysis of models with intrinsic variability in their parameters. Sources of noise in biology are ubiquitous, complex, and often neglected or at least under-appreciated in systems biology analyses of model identifiability and parameter inference. This is thus an important and timely contribution to the literature that will be widely useful. The inclusion of open-source code in Julia is an additional strength of the manuscript. I have only minor requests/suggestions for possible improvement to the paper. These are given below.

1. While the analysis of how skewness or bimodality affect identifiability is interesting, at least in my experience, a far more common occurrence in biological modeling is the choice of a prior that is particularly uninformative (e.g. uniform over large interval). Would such a prior choice affect the inference/identifiability results, and how? This would be a useful example.

2. I appreciate the advantages of & the need to strive for model simplicity, however, practically, two species is really a lower bound for realistic model sizes in sys bio. I think it would be really beneficial if an example (or even just some discussion without an example) of a larger (e.g. 3 species) model, in light of the analyses contained in this work: studying identifiability with random parameters & the impact of prior choice, etc.

3. In Fig 9, please could you improve the caption/description: it is a figure containing quite a lot of details and it is currently hard to figure out several details. Are colors indep MCMC chains? The different dashed/dotted lines in A are hard to distinguish. I cannot see orange solid lines in B. Also what does grey represent in B?

**Have the authors made all data and (if applicable) computational code underlying the findings in their manuscript fully available?**

Reviewer #1: Yes

Reviewer #2: Yes

Reviewer #3: Yes

PLOS authors have the option to publish the peer review history of their article (what does this mean?). If published, this will include your full peer review and any attached files.

Reviewer #1: No

Reviewer #2: No

Reviewer #3: No

Figure Files:

Data Requirements:

Reproducibility:

References:

---

## [Decision Letter · Decision Letter 1]

14 Nov 2022

Dear Professor Simpson,

We are pleased to inform you that your manuscript 'Efficient inference and identifiability analysis for differential equation models with random parameters' has been provisionally accepted for publication in PLOS Computational Biology.

Best regards,

Ricardo Martinez-Garcia

Academic Editor

PLOS Computational Biology

Jason Papin

Editor-in-Chief

PLOS Computational Biology

Reviewer's Responses to Questions

**Comments to the Authors:**

Reviewer #1: The authors addressed all of my comments and concerns. Thank you!

Reviewer #2: The authors have satisfyingly addressed all of my concerns, so I would recommend the manuscript for publication.

Reviewer #3: In this revision the authors have addressed all of my previous concerns.

**Have the authors made all data and (if applicable) computational code underlying the findings in their manuscript fully available?**

Reviewer #1: Yes

Reviewer #2: Yes

Reviewer #3: Yes

PLOS authors have the option to publish the peer review history of their article (what does this mean?). If published, this will include your full peer review and any attached files.

Reviewer #1: No

Reviewer #2: No

Reviewer #3: No

---

## [Editor Report · Acceptance letter]

21 Nov 2022

PCOMPBIOL-D-22-01128R1 

Efficient inference and identifiability analysis for differential equation models with random parameters

Dear Dr Simpson,

I am pleased to inform you that your manuscript has been formally accepted for publication in PLOS Computational Biology. Your manuscript is now with our production department and you will be notified of the publication date in due course.

With kind regards,

Anita Estes
